# Thickness of the divide and flank of the West Antarctic Ice Sheet through the last deglaciation

Perry Spector[1], John Stone[2], and Brent Goehring[3]

[1]Berkeley Geochronology Center, 2455 Ridge Road, Berkeley, CA, USA
[2]Department of Earth and Space Sciences, University of Washington, Seattle, WA, USA
[2]Department of Earth and Environmental Sciences, Tulane University, New Orleans, LA, USA

**Correspondence:** Perry Spector (pspector@bgc.org)

**Abstract.** We report cosmogenic-nuclide measurements from two isolated groups of nunataks in West Antarctica: the Pirrit Hills, located midway between the grounding line and the divide in the Weddell Sea sector, and the Whitmore Mountains, located along the Ross-Weddell divide. At the Pirrit Hills, evidence of glacial-stage ice cover extends $\sim$320 m above the present ice surface. Subsequent thinning mostly occurred after $\sim$14 kyr B.P., and modern ice levels were established some time after $\sim$4 kyr B.P. We infer that, like at other flank sites, these changes were primarily controlled by the position of the grounding-line downstream. At the Whitmore Mountains, cosmogenic $^{14}$C concentrations in bedrock surfaces demonstrate that ice there was no more than $\sim$190 m thicker than present during the past $\sim$30 kyr. Combined with other constraints from West Antarctica, the $^{14}$C data imply that the divide was thicker than present for a period of less than $\sim$8 kyr within the past $\sim$15 kyr. These results are consistent with the hypothesis that the divide initially thickened due to the deglacial rise in snowfall, and subsequently thinned in response to retreat of the ice-sheet margin. We use these data to evaluate several recently-published ice-sheet models at the Pirrit Hills and Whitmore Mountains. Most of the models we consider do not match the observed timing and/or magnitude of thickness change at these sites. However, one model performs relatively well at both sites, which may, in part, be due to the fact that it was calibrated with geological observations of ice thickness change from other sites in Antarctica.

## 1 Introduction

Our knowledge of past thickness changes of the West Antarctic Ice Sheet (WAIS) is largely derived from geologic evidence collected from the continental shelf seafloor and from sites near the margin of the present-day ice sheet. Less is known about changes in the high-elevation WAIS interior where outcropping mountains and thus geologic evidence are sparse. The only geological constraints come from exposure dating at the Ohio Range and Mt. Waesche (Fig. 1) (Ackert et al., 1999, 2007, 2013). In this paper we describe glacial-geologic observations and cosmogenic-nuclide measurements from two isolated nunatak groups that rise through the WAIS: the Pirrit Hills located midway between the grounding line and the divide, and the Whitmore Mountains, which lie along the divide between the Weddell and Ross Sea sectors (Fig. 1). These data provide information about the magnitude and timing of thickness changes in interior West Antarctica since the Last Glacial Maximum (LGM), which, in turn, help to identify the glaciological processes that were most important for WAIS thickness over this time period.

At the end of the last ice age in West Antarctica, the three processes that likely exerted the greatest influence on ice thickness were (i) the retreat of the margin, (ii) the increase of the accumulation rate, and (iii) warming of the ice surface (Cuffey and Patterson, 2010, Chapter 11.4.2). Margin retreat is expected to have propagated a diffusive wave of thinning upstream, with the greatest and earliest thinning near the coast, and the least and most delayed thinning at the divide. This dynamic thinning would have been opposed by the increase in the accumulation rate, which, at the WAIS Divide ice-core site (Fig. 1), approximately doubled between 18 and 15 kyr B.P. (Fudge et al., 2016). Although the accumulation-rate increase was likely, in part, a result of higher atmospheric temperatures, Fudge et al. (2016) demonstrate that other factors, which may have included reduced sea ice or changes in atmospheric circulation, were also important. The deglacial warming would also have softened the ice sheet and allowed it to flow faster, thereby inducing thinning. However, while the response of ice thickness to retreat of the margin and to increased snowfall is immediate, the response to surface warming is minimal until the base of the ice sheet warms, which, for West Antarctica, is expected to require roughly 10 to 30 kyr (Whillans, 1981; Cuffey and Patterson, 2010, Chapter 11.4.2). Because surface temperatures remained relatively low until ∼15 kyr B.P. (Cuffey et al., 2016), any thinning due to surface warming was likely limited to the late Holocene.

The combined effect of these processes can result in a complex history of ice thickness change at a given site, with thickening and thinning potentially both occurring over the course of the deglaciation as the balance between the different processes shifts (e.g., Alley and Whillans, 1984; Cuffey and Clow, 1997; Steig et al., 2001; Hall et al., 2015). At multiple sites arrayed along a flowline, the ice-thickness history of each site will vary depending, among other things, on whether the site is located closer to the divide or to the grounding line.

Exposure ages from nunataks near the present-day WAIS margin indicate progressive surface lowering as the grounding line neared (e.g., Johnson et al., 2014; Hein et al., 2016; Spector et al., 2017), implying that dynamic thinning was the dominant process on the lower flanks of the ice sheet. We have little knowledge as to whether any thinning due to ice-sheet warming has occurred in West Antarctica. The only relevant constraints we are aware of are exposure ages from lower Reedy Glacier (Fig. 1) that demonstrate that ice levels there have remained stable since ∼7 kyr B.P. (Spector et al., 2017; Todd et al., 2010), which suggests that surface warming has not yet induced significant thinning. Constraints on thickness changes from the WAIS interior are also scarce (Ackert et al., 1999, 2007, 2013), and thus the relative importance of different glaciological processes near the divide remains poorly known. It has been hypothesized that, in contrast to the monotonic thinning near the coast, the divide region initially thickened in response to the deglacial rise in snowfall and subsequently thinned only once the dynamic effects of downstream retreat began to outpace the increased snowfall (Steig et al., 2001; Cuffey et al., 2016).

The data we report in this paper are consistent with this hypothesis. We find that at the Pirrit Hills, the WAIS stood at a highstand early in the deglaciation and thinned monotonically through the Holocene, similar to previously published records from sites near the ice-sheet margin. In contrast, at the Whitmore Mountains the WAIS appears to have (i) been no thicker than present, and possibly thinner, during the LGM when snowfall rates were lowest, and (ii) reached a highstand sometime in the last ∼15 kyr, once accumulation rates had climbed from their LGM low. Because these data are some of the only constraints on past ice thickness from the WAIS interior, they are particularly valuable for evaluating the performance of ice-sheet model simulations of the last deglaciation, and we provide an example of this using several recently published simulations.

## 2 Field sites and glacial geology

### 2.1 Pirrit Hills

The Pirrit Hills emerge from the WAIS at an elevation of ∼1300 m from a slow-flowing portion of the Institute Ice Stream catchment in the Weddell Sea sector, ∼200 km from both the grounding line and from the divide. The main massif of the Pirrit Hills is composed of granite, although some of the small satellite nunataks are metasedimentary. The massif consists of a few major and several minor peaks, linked by catenary arêtes and buttressed by steep spurs. The spurs divide cirque basins whose floors are concealed by the present-day ice sheet. Katabatic winds flow from SW to NE over these mountains and have deposited a ramp of snow on their upwind side that rises over a distance of 5-10 km to the saddle between Mt. Tidd and Mt. Turcotte (Fig. 2a). Northeast of here, where the winds are forced to descend, warm, and become turbulent, there is a 1-2 km wide blue-ice ablation zone, which sits 600 m below the saddle. Englacial debris has accumulated here in sheets and moraine ridges that onlap the base of the mountains. Above the level of the modern ice sheet, isolated glacial deposits occur on the narrow bedrock ridges of Mts. Axtell, Tidd, and Turcotte above the modern ablation zone. The highest deposits we found were on the NE buttresses of Mt. Axtell and Mt. Tidd at ∼320 m and ∼340 m above the modern ice surface, respectively (Figs. 2a, 3a, 4a). The density of boulders and cobbles is considerably higher at these altitudes than lower on the two ridges, with debris covering most of the area where accumulation is possible. For example, compare Figure S1 to Figure S2. The deposits are typically lightly weathered and consist of granite lithologies that outcrop at the Pirrit Hills. In order to identify cobbles and boulders as having been glacially transported, we searched for rounded to sub-angular clasts bearing impact marks and blunted corners. The underlying bedrock is more oxidized than the glacial debris and displays very little evidence of glacial erosion (e.g. Figures S1 and S3). Uphill from the depositional limit, the intensity of bedrock weathering increases. Some of the most heavily-weathered bedrock surfaces at the Pirrit Hills were found above the depositional limit, on the bench near Mt. Axtell's summit (Fig. 3a). The granite here exhibits gritty exfoliation sheets, case hardening, and delicate tafoni. Grus and felsenmeer has accumulated in low-lying areas between rounded joint-bound bedrock blocks.

Although the Pirrit Hills were carved by mountain glaciers, this is a relic alpine landscape unrelated to the present-day or Pleistocene WAIS. The glacial deposits establish that the ice sheet here was at least 320-340 m thicker than present at least once in the past. There is no evidence that ice has reached above the depositional limit, and the absence of any glacial debris on the Axtell bench, along with the difference in bedrock weathering between the bench and the ridge below, suggests that ice has not been more than ∼450 m thicker than present. The observation that glacially-transported cobbles and boulders are less weathered than the bedrock on which they rest indicates that past ice cover was largely frozen at its base, which is consistent with the limited expression of glacial erosion features or well-rounded glacial debris. The greater abundance of debris near the depositional limit suggests that the ice sheet stood at or near its highstand for longer than it did at lower levels.

### 2.2 Mt. Seelig, Whitmore Mountains

The northwest ridge of Mt. Seelig (the only site we visited in the Whitmore Mountains) divides two partially-buried cirques and climbs from the ice-sheet surface at ∼2200 m to a local ice cap that drapes the mountain top (Fig. 2b). Easterly winds

flowing toward the Ross Sea maintain a blue-ice area alee of the ridge. These winds have deposited small snowfields on the upper, wider portion of the ridge and drifts behind the lower, narrow ridge (Figs. 2b, 3c, 4b). To minimize the likelihood of sampling from areas prone to snow cover, we kept to the windswept edge of the ridge overlooking the steep headwall shown in Figures 3c and 4b. Bedrock surfaces here display subaerial weathering features including oxidation, granular disintegration, and weathering pits (Fig. 4b). We observed no evidence of recent glacial erosion.

Unlike the Pirrit Hills, we found no glacially-transported cobbles or boulders perched on bedrock surfaces. Glacially transported rock is less likely here at the divide because there is little area from which to source debris. The only glacial deposit we found was a small patch (several square meters) of indurated and weathered till ~150 m above the modern ice surface (Fig. S4). The till consists of poorly-sorted granite clasts and a few striated metasedimentary rocks embedded in a fine-grained matrix. These characteristics indicate deposition by wet-based ice. The simplest interpretation is that the patch of weathered till is a remnant of a more extensive ancient deposit that has been largely eroded away. Therefore, to summarize, we find no geomorphic evidence in support of higher ice levels at Mt. Seelig during or since the LGM.

## 3  Methods

### 3.1  Sample collection

At the Pirrit Hills, we sampled elevation transects of glacial deposits to determine the age and height of the most recent highstand and to chronicle the subsequent thinning (Figs. 3a-b, S5). Cosmogenic [10]Be has accumulated in these samples since their initial exposure in the ablation zones on the northeast side of the mountains. Anomalously young exposure ages can result from post-depositional erosion of the rock surface or shielding by snow or till. To minimize the possibility of post-depositional shielding, we sampled isolated deposits resting in stable positions on narrow, windswept bedrock ridges (e.g. Figures 4a, S3). To minimize the likelihood of erosion, we preferentially selected lightly-weathered rocks retaining evidence of glacial modification.

On the northwest ridge of Mt. Seelig, where recent glacial deposits are absent, we collected an elevation transect of bedrock samples from stable surfaces to identify past highstands and compare exposure and ice cover at different altitudes. Despite targeting sites unlikely to have been snow covered in the past, in places the only exposed bedrock was located within meters of snowfields or the summit ice cap (e.g. Fig. 4c), and these samples may have been covered in the past.

At both sites, we measured sample elevations using drift-corrected barometric measurements, calibrated with geodetic GPS measurements. Accuracy is estimated to be $\pm$ 3-4 m based on repeat measurements. Elevations are reported relative to the EGM96 geoid. At the Pirrit Hills, we determined the elevations of the modern blue-ice areas at the bases of Mts. Axtell, Tidd, and Turcotte, which are needed to calculate sample heights above the ice, using a high-resolution digital elevation model with < 1 m vertical uncertainty in this region (Howat et al., 2019). For all samples, we measured the topographic shielding from the cosmic-ray flux from vertically-oriented fisheye photographs.

## 3.2 Analysis strategy for bedrock samples from Mt. Seelig

To determine the history of exposure and ice cover on million-year timescales, we measured the long-lived cosmogenic nuclides $^{26}$Al, $^{10}$Be, and $^{21}$Ne in quartz, which will be described in a forthcoming publication. Here we describe measurements of cosmogenic $^{14}$C in quartz. Because $^{14}$C decays quickly (5.7 kyr half-life), its concentration is only sensitive to exposure and ice cover that occurred in the past ~30-35 kyr (Goehring et al., 2019a). After ~30-35 kyr of continuous exposure, a sample will be saturated with respect to $^{14}$C, at which point nuclide production is balanced by decay, and the concentration is no longer time dependent. Therefore, unlike longer-lived cosmogenic nuclides, $^{14}$C has no memory of exposure or ice cover that occurred prior to ~30-35 kyr ago.

## 3.3 Analytical methods

Quartz was separated from crushed rock samples, sieved to 0.25-0.5 mm, and purified using surfactant separation, flotation in heavy liquids, and repeated etching in 2 % HF. Beryllium was extracted from quartz aliquots at the University of Washington Cosmogenic Nuclide Lab by addition of $^{9}$Be carrier, dissolution in HF, ion exchange chromatography, and precipitation of Be hydroxide (Ditchburn and Whitehead, 1994). Measurements of total Be by inductively coupled plasma optical emission spectrometry (ICP-OES) on aliquots taken after sample dissolution indicate that purified quartz separates were contaminated with 0.001-0.007 % beryl (and/or other Be-bearing minerals) not separated by the procedure described above. Therefore, we calculated $^{10}$Be concentrations using the ICP-OES determinations of total Be rather than the amount of Be added as carrier. Beryllium isotope ratios were measured at the Lawrence Livermore National Laboratory Center for Accelerator Mass Spectrometry (LLNL CAMS). Beryllium isotope ratios were measured relative to the ICN 01-5-4 standard, and $^{10}$Be concentrations given in Table S1 are calculated relative to a value of $^{10}$Be/$^{9}$Be $= 2.851 \times 10^{-12}$ (Nishiizumi et al., 2007). The batch of samples from Mt. Axtell and Mt. Tidd produced a process blank of $5{,}000 \pm 900$ atoms $^{10}$Be. The batch from Mt Turcotte gave an unusually high process blank of $133{,}000 \pm 8{,}000$ atoms $^{10}$Be, which is likely due to inadvertent cross-contamination from one of the accompanying samples. We have based the blank subtraction for these samples on this value, however it may be more appropriate to use a more typical laboratory average of ~10,000 atoms. This choice turns out to be unimportant because all of the samples from Mt. Turcotte are pre-exposed (see Section 4.1 below). For samples whose exposure ages define the deglaciation history discussed below, blank subtractions amount to 0.1-0.7 % of the total $^{10}$Be.

Uncertainties in $^{10}$Be concentrations (Table S1) include all known sources of laboratory uncertainty combined in quadrature. For samples prefixed "13-NTK", we assigned larger uncertainties to Be isotope ratios than those reported by LLNL CAMS. The larger uncertainties are based on analyses of isotope ratio standards (Nishiizumi et al., 2007) prepared at the University of Washington (UW) and analyzed along with samples. During the period 2011-2015 these UW standards commonly scattered in isotope ratio by more than internal accelerator standards. To assess this additional source of isotope-ratio error we determined the additional percentage error required to bring the chi-squared of the UW standards to 1.0. In the case of the 13-NTK samples, this additional error of 4.8 %, based on 7 standard analyses, has been added in quadrature to the isotope ratio error reported by

LLNL CAMS. Bias in the ratios of the UW standards run with the 13-NTK samples was -2.4 $\pm$ 3.9 %, and because the scatter exceeds the bias, no correction has been applied to the isotope ratios and resulting concentrations in Table S1.

Quartz aliquots for $^{14}$C measurement were twice etched in a 5 % HF and 5 % $HNO_3$ solution on a shaker table, each time for 24 hours, then twice in a 1 % HF and 1 % $HNO_3$ solution in a 50°C ultrasonic bath, each time for 24 hours. Experiments at Tulane University show that this removes potential organic contaminants from the surfactant separation (Nichols and Goehring, 2019). Carbon was extracted from quartz using the Tulane University Carbon Extraction and Graphitization System (Goehring et al., 2019b). This entails fusion of quartz in vacuo using a $LiBO_2$ flux, cryogenic and redox collection and purification of $CO_2$, and manometric measurement of $CO_2$ yield. Carbon isotope ratios were measured at the National Ocean Sciences Accelerator Mass Spectrometry facility. The total process blank of 6.47 $\pm$ 0.68 $\times 10^4$ $^{14}$C atoms is based on the long-term average of blanks and represents approximately 1-2 % of the total $^{14}$C atoms measured in the samples. Measurements of $\delta^{13}$C were made at the Stable Isotope Facility at the University of California, Davis.

Repeat measurements of the CRONUS-A quartz standard (Jull et al., 2015) at Tulane University scatter by 5.2 %. This is higher than expected from analytical uncertainties for most samples. The samples from Mt. Seelig all have analytical uncertainties less than 1.5 %, and thus we adopt a uniform uncertainty of 5.2 % for all samples. Results of $^{10}$Be and $^{14}$C measurements are reported in the supplementary information as well as online in the ICE-D: ANTARCTICA database located at http://antarctica.ice-d.org.

Some replicate $^{14}$C measurements on samples from elsewhere in Antarctica that were prepared at Tulane University scatter significantly more than expected from their analytical uncertainty alone (e.g. Nichols et al., 2019). Because of the possibility of contamination from modern carbon, it is easier to measure an erroneously high $^{14}$C concentration than an erroneously low concentration. Analyses that result in erroneously low $^{14}$C concentrations can typically be identified by automated monitoring of the C extraction process (Goehring et al., 2019b). None of the samples from Mt. Seelig either (i) produced $^{14}$C concentrations in excess of predicted saturation values or (ii) indicated incomplete C extraction, and therefore we have no reason to believe that these measurements have unrecognized analytical error. The conclusions that we draw about thickness changes at the ice-sheet divide are subject to the accuracy of these data. Refer to the interactive online discussion of this article for a more detailed discussion of this issue.

## 3.4 Production rates of $^{10}$Be and $^{14}$C

We compute production rates for $^{10}$Be and $^{14}$C in quartz using the "LSDn" production-rate scaling method (Lifton et al., 2014), as implemented in version 3 of the online exposure calculator described by Balco et al. (2008) and subsequently updated. Beryllium-10 production rates by spallation are based on the CRONUS-Earth "primary" production-rate calibration dataset (Borchers et al., 2016). Carbon-14 production rates are calibrated using repeat measurements of CRONUS-A at Tulane University. CRONUS-A was collected from a slowly-eroding site in the McMurdo Dry Valleys (elevation: 1679 m; distance from Whitmore Mountains: 1650 km) that remained ice free during the LGM. CRONUS-A is therefore assumed to be saturated with respect to $^{14}$C (Jull et al., 2015). Calibrating production rates in this way minimizes uncertainties associated with

scaling production rates from sites at lower latitudes. We assume that LSDn scaling is accurate over the ∼1 km elevation range between CRONUS-A and our highest samples from the Whitmore Mountains.

## 4 Results

### 4.1 WAIS thinning history at the Pirrit Hills

Glacial deposits at the Pirrit Hills have apparent exposure ages that range from 1 Myr to ∼4 kyr (Fig. 5a). 11 of the 18 samples have ages greater than 60 kyr, while the remaining samples are all younger than 18 kyr. Qualitatively, there appears to be a relationship between exposure-age and the degree of rock weathering (see, for example, Figure S6). Because, as discussed above, our sampling considerations minimize the possibility of post-depositional (i) cover by snow or till, (ii) disturbance, or (iii) erosion, all of which cause anomalously young ages, we interpret the youngest ages as dating deposition during or

following the LGM and the older ages as the result of prior cosmic-ray exposure. Strictly, we cannot rule out the possibility that the youngest ages also record minor prior exposure. However, the similarity of these ages to the youngest ages from the nearby Heritage Range (discussed below) suggests that this effect is either absent or minor, amounting to less than ∼1-2 kyr.

   The exposure age of a cobble from the depositional limit on Mt. Axtell, ∼320 m above the modern ice surface, indicates that ice reached its highstand by 18 ± 1.0 kyr B.P. A boulder sampled ∼15 m below the limit has an age of 14.4 ± 0.8 kyr

B.P., demonstrating that ice levels persisted near the highstand for at least 3-4 kyr, which is consistent with the abundance of debris near the depositional limit. Because the ice surface varied slowly at this height, it is possible that samples may have been exposed in the ablation zone for centuries or millennia prior to physical deposition on bedrock (Ackert et al., 2011).

   Below this level, deposits are more sparse (Figs. S1 and S2), suggesting that thinning from the highstand occurred relatively rapidly and that samples were exposed in the ablation zone only briefly before being deposited. The thinning is constrained

by only two samples from Mt. Tidd (Fig. 5a); other samples from Mts. Axtell and Turcotte are pre-exposed. By 6.9 ± 0.4 kyr B.P., the ice surface had lowered ∼140 m from the highstand. Another 110 m of thinning occurred in the subsequent ∼2.7 kyr, bringing ice levels to within ∼60 m of the modern surface by ∼4.2 kyr B.P.

   This result is similar to thinning chronologies from the Heritage Range and from the Pensacola Mountains, sites in the Weddell Sea sector that are more seaward than the Pirrit Hills (Fig. 1). At the Heritage Range, located in the southern Ellsworth

Mountains, a highstand 250-500 m above the modern ice surface was reached by ∼18 kyr B.P. and thinning to the modern ice level occurred ∼6-3 kyr B.P. (Hein et al., 2016; Sugden et al., 2017; Bentley et al., 2010) (previously-published ages have been recalculated to be consistent with data presented here). At the Williams and Thomas Hills on the west side of the Pensacola Mountains, ice thinned at least 500 m between 11 and 4 kyr B.P. (Balco et al., 2016; Bentley et al., 2017). Although initial exposure dating from a third site in the Pensacola Mountains, the Schmidt Hills, found no evidence for thicker ice in the past

100 kyr (Balco et al., 2016; Bentley et al., 2017), recent cosmogenic [14]C measurements demonstrate that ice was at least 800 m thicker during the LGM (Nichols et al., 2019). Therefore, to summarize, exposure-dating results from the Pirrit Hills, Heritage Range, and Pensacola Mountains suggests that the timing of thinning during the last deglaciation was relatively similar across the lower flank of the Weddell Sea sector of the WAIS.

## 4.2 Upper limit on the highstand of the WAIS at Mt. Seelig

At Mt. Seelig, four bedrock samples have [14]C concentrations indistinguishable from saturation (Fig. 5b) and must have been continuously exposed for at least the past ~30 kyr. The elevation of the lowest saturated sample places an upper limit of ~190 m on the highstand of the Ross-Weddell divide, relative to its present altitude, over the past ~30 kyr. The remaining six bedrock samples have [14]C concentrations below saturation and were therefore shielded from the cosmic-ray flux for some portion of the past ~30 kyr. Of these, three were collected more than 190 m above the modern ice surface, which eliminates the possibility that they could have been covered by the WAIS. Because it is very unlikely that these samples experienced significant surface erosion or till cover (refer to Sections 2 and 3), processes that would reduce [14]C concentrations, we interpret these samples to have been covered by expanded snow fields or the summit ice cap for some portion of the past ~30 kyr. Only thin snow cover would be required to block the majority of the cosmic-ray flux. If we assume snow with an average density of 500 kg m$^{-3}$, a typical value for near-surface snow and firn in central West Antarctica (Mayewski et al., 2005), the [14]C production rate would be reduced by 75 % beneath 2 m of snow and by more than 90 % beneath 4 m. A lower limit on the duration of snow cover can be obtained by assuming (i) very thick snow, in which the [14]C production rate in the underlying bedrock approaches zero, and (ii) that snow cover occurred very recently. With these assumptions, the [14]C concentrations require cover for at least 1-2 kyr.

The other three samples with [14]C concentrations below saturation are more ambiguous. Because they were collected below 190 m, they are consistent with having been covered by a thicker WAIS. However, we cannot exclude the possibility that some or all of the shielding these samples require came from snow cover. These samples are discussed in more detail in the next section.

While snow cover is the only simple explanation for [14]C concentrations below saturation in samples collected above 190 m, there is not an obvious relationship between proximity to present-day snowfields and whether a sample is [14]C saturated or not. Although all samples from Mt. Seelig are estimated to have been collected within approximately 5-20 m of snowfields, some, such as those from 236 and 295 m above the modern ice level, were collected from prominent outcrops along the cliff edge, and it is more difficult to envision that these sites were covered by thick snowfields. The exception to this is the highest sample from Mt. Seelig, which has a [14]C concentration below saturation and was collected from a very small outcrop (few square meters) that barely protrudes through the margin of the summit ice cap (Fig. 4c). How susceptible each sample site is to snow cover is likely related to local wind conditions near the cliff edge, which may have changed since the LGM.

## 4.3 Ice cover and exposure scenarios for the three lowest elevation bedrock samples from Mt. Seelig

As discussed above, the [14]C concentrations of the three lowest elevation samples from Mt. Seelig are compatible with multiple different scenarios of exposure and of cover by the WAIS and/or local snow fields. To explore the range of possible scenarios we consider a model consisting of the following three stages: (i) initial exposure of the sample for sufficient time (>~30 kyr) that it is [14]C saturated, (ii) subsequent ice cover by a thicker WAIS during which [14]C is lost to decay, and (iii) a final period of exposure that begins sometime in the past 30 kyr and continues to the present. We do not consider scenarios with more than three stages because the response time of ice thickness changes at the divide to changes in accumulation rate or to the position

of the margin is millennial (Cuffey and Patterson, 2010, Chapter 11.4.2), and thus high-frequency thickness fluctuations are unlikely. We assume that burial and re-exposure are immediate and that samples are buried by a sufficient thickness of ice or firn to completely halt production. We also assume that no subglacial or subaerial erosion occurs, which is supported by geomorphic observations (see Section 2.2). At the end of a three-stage scenario, the present-day $^{14}$C concentration, $N$, is given
by the following equation:

$$N = \frac{P}{\lambda}\Big[1 + \exp(-\lambda t_{cover}) - \exp(-\lambda t_{expose})\Big] \tag{1}$$

where $P$ is the total production rate from spallation and from muon interactions, $\lambda$ is the decay constant, $t_{cover}$ is the time of initial cover by the WAIS, and $t_{expose}$ is the time of subsequent re-exposure.

The results of this calculation can be explained graphically by considering a diagram like Figure 6a that has axes of $t_{cover}$
and $t_{expose}$. For a given $^{14}$C concentration, there exists a set of possible ice-cover and exposure histories $\{t_{cover}, t_{expose}\}$ that define a curve. The curve begins at some point where $t_{expose}$ is equal to zero and, at high values of $t_{cover}$, approaches a maximum $t_{expose}$ equal to what is commonly referred to as the "simple exposure age", that is, the exposure duration implied by the $^{14}$C concentration assuming only one period of exposure. Uncertainty in the measured $^{14}$C concentration widens the curve into a band. Although a separate band can be computed for each of the samples from below ∼190 m, for the sake of
simplicity we have combined the bands of the two lowest elevation samples in Figure 6a because they were collected at similar elevations and have similar $^{14}$C concentrations (Fig. 5b). The gray band labeled "zero snow cover" therefore represents the set of ice-cover and exposure histories permitted by this pair of samples. The area in white to the left of this band represents the set of $\{t_{cover}, t_{expose}\}$ pairs that underpredict the observed $^{14}$C concentrations and are thus forbidden. As discussed above, the $^{14}$C concentrations of these samples require some combination of cover by the WAIS and/or by expanded local snow fields.
Therefore, the gray area labeled "partial snow cover" represents the set of permissible $\{t_{cover}, t_{expose}\}$ pairs assuming that some portion of the required cover was due to snowfields. The end-member scenario that the samples were never covered by the WAIS and that all cover was due to snowfields is represented by the 1:1 line. An equivalent diagram for the sample from ∼130 m above the ice surface is shown in Figure 7d.

The $^{14}$C concentrations of the two samples collected near the modern ice surface would correspond to simple exposure ages
of ∼10-15 kyr, under the assumption that they were previously ice-covered for a sufficient amount of time to remove any pre-existing $^{14}$C. This is represented by the asymptote of the "zero snow cover" band in Figure 6a. However, under the three-stage model of Equation 1, this pair of samples could have also been exposed by WAIS thinning (i) within the past ∼10 kyr, with more recent re-exposure requiring relatively brief prior ice cover, or (ii) prior to ∼15 kyr B.P., given the possibility of cover by local snow fields (Fig. 6a). The primary assumption of our model is that these samples were initially $^{14}$C saturated. Relaxing
this assumption by considering scenarios with initial $^{14}$C concentrations below saturation shifts the gray band in Figure 6a labeled "zero snow cover" down and to the right (i.e., towards briefer and more recent WAIS cover). Therefore, because we also consider scenarios of partial snow cover, our assumption of initial $^{14}$C saturation has the effect of maximizing the number of exposure and WAIS-cover scenarios that are permitted by the observed $^{14}$C concentrations.

## 5 Discussion

### 5.1 Competition between snowfall and dynamic thinning at the divide

At the WAIS Divide ice-core site, the accumulation rate was lowest during the LGM and then doubled to near-modern values between 18 and 15 kyr B.P. (Figure 6c; Fudge et al., 2016). Although the magnitude of the accumulation-rate increase at the
Pirrit Hills and the Whitmore Mountains (located 530 km and 360 km away, respectively) may not have been the same as at WAIS Divide, the timing of changes was probably similar because (i) all three sites are fed by storms originating in the Amundsen Sea low (Hogan, 1997; Turner et al., 2013), and (ii) the accumulation rate increased considerably in both East and West Antarctica at this time (Bazin et al., 2013; Veres et al., 2013).

At the Pirrit Hills, ice levels appear to have lowered monotonically following the LGM (Fig. 5a) despite the deglacial
increase in snowfall, implying that the dominant glaciological process was thinning induced by retreat of the grounding line downstream. The same interpretation is implied by thinning records from the Heritage Range and the Pensacola Mountains (Hein et al., 2016; Balco et al., 2016; Bentley et al., 2010, 2017; Nichols et al., 2019), as well as other sites located near the present-day ice-sheet margin (e.g. Johnson et al., 2014; Stone et al., 2003; Spector et al., 2017). We note that, as discussed in Section 1, thinning due to surface warming (and the eventual increase of ice deformation rates near the bed) is not expected
to have occurred prior to the late Holocene at the earliest, by which time the majority of thinning to the modern ice level was complete at the Pirrit Hills and most other West Antarctic sites. As discussed by Hein et al. (2016) in regard to the Heritage Range, the onset of thinning at the Pirrit Hills (and other sites) may have been delayed by the deglacial snowfall increase. Below we show that, in contrast to the monotonic thinning near the margin, the divide appears to have initially thickened following the LGM due to the increased snowfall, and only thinned once the dynamic effects of margin retreat began to outpace the
thickening from snowfall.

Figure 6a shows that if we know when the two lowest elevation samples from Mt. Seelig were most recently exposed by WAIS thinning, then we can place a constraint on when they were initially buried by thickening. Because this pair of samples was collected very close to the modern ice surface, the onset of their burial and subsequent re-exposure are nearly equivalent to when the divide here became thicker than present and when it thinned to its modern level. However, for this to be meaningful,
we must first establish that the divide actually was thicker than present. Although the flanks of the WAIS were certainly thicker during the LGM, the divide is where the smallest thickness changes are expected over glacial-interglacial cycles, and, as discussed in Section 4, the $^{14}$C concentrations alone do not require cover by a thicker ice sheet. The strongest evidence for thicker ice at the Whitmore Mountains comes from the Ohio Range in the southern Transantarctic Mountains (Fig. 1), where the WAIS was at a highstand ∼125 m above present between 12 and 9 kyr ago (Ackert et al., 2007, Figure 6b). The inference
of thicker ice at the Whitmore Mountains is further supported by analysis of $\delta^{18}$O records from the Byrd and Taylor Dome ice cores, which imply that the ice surface at Byrd Station, located in the upper portion of the Ross Sea catchment, lowered ∼250 m during the Holocene (Steig et al., 2001).

These considerations strongly suggest that the less-than-saturated $^{14}$C concentrations in the lowest two bedrock samples from Mt. Seelig are partially or fully attributable to cover by a thicker WAIS. These samples would have emerged only after sites

downstream had thinned to present-day ice levels because dynamic thinning propagates upstream from the ice-sheet margin. In all sectors of the WAIS, present-day ice levels were reached in the mid-late Holocene (Figs. 5a and 6d; Stone et al., 2003; Todd et al., 2010; Johnson et al., 2014; Balco et al., 2016; Bentley et al., 2017; Hein et al., 2016; Spector et al., 2017); however, the most relevant sites are those nearest the flowlines that descend from the Whitmore Mountains. In the Weddell Sea sector, these

are the Pirrit Hills and Heritage Range, where, as described in Section 4.1, present-day ice levels were reached after 4-5 kyr B.P. (Fig. 6d). In the Ross Sea sector, lower Reedy Glacier is the most relevant site. Exposure dating here demonstrates that thinning coincided with deglaciation of a large portion of the Ross Sea 9-7 kyr ago (Spector et al., 2017; Todd et al., 2010). By ∼7-6 kyr B.P., most of the thinning was complete; the ice sheet stood within ∼50 m of the present-day surface, down from a highstand that was at least ∼150 m above present and likely 200-250 m or higher based on the height of depositional limits

farther upstream on Reedy Glacier (Todd et al., 2010).

Thinning to the modern ice level at Mt. Seelig therefore could not have occurred before 7 kyr ago (i.e., before modern ice levels were reached on lower Reedy Glacier). If the two lowest samples emerged 7 kyr ago, their [14]C concentrations require initial burial sometime after ∼15 kyr B.P. (Fig. 6a), which places an upper limit of 8 kyr on the duration of thicker-than-present ice cover. More recent emergence would require more recent burial and a shorter burial duration. The finding of brief and recent

ice cover (≤ 8 kyr cover within the past ∼15 kyr) is insensitive to the primary assumption in Figure 6a that these samples were [14]C-saturated prior to being ice covered. Relaxing this assumption would actually imply that the onset and duration of burial were later and more brief, respectively, than implied by Figure 6a. Although ice cover may have been relatively recent, such as scenario 'e' in Figure 6, histories with earlier cover, such as scenario 'f', are more consistent with the timing of the highstand at the Ohio Range (Fig. 6b; Ackert et al., 2007, 2013). If true, a scenario like 'f' implies that modern ice levels were reached at

the divide earlier than at flank sites in the Weddell Sea sector (e.g. Pirrit Hills), which would likely have been a consequence of the fact that deglaciation of much of the southern Ross Sea was complete by ∼7 kyr B.P. (Spector et al., 2017; Todd et al., 2010), while grounding-line changes in the Weddell Sea sector appear to have continued into the late Holocene (Fig. 5a; Siegert et al., 2013; Hein et al., 2016; Nichols et al., 2019; Johnson et al., 2019; Siegert et al., 2019).

These constraints demonstrate that the WAIS at the Whitmore Mountains was the same thickness or thinner than present

prior to the most recent highstand, and that this highstand was reached sometime in the last ∼15 kyr. This result is consistent with the hypothesis described by Steig et al. (2001) and more recently by Cuffey et al. (2016) that the divide thickened early in the deglaciation due to the rise in snowfall and subsequently thinned only once the dynamic effects of margin retreat began to outpace the thickening from snowfall. As noted above, any thinning induced by surface warming is expected to have been delayed until the late Holocene at the earliest and thus does not affect our findings. We note that our results are not necessarily

representative of thickness changes beyond the Ross-Weddell divide. This is because other divide segments (e.g. Weddell-Amundsen and Amundsen-Ross; Fig. 1) likely experienced somewhat different histories of snowfall and dynamic thinning induced by grounding-line retreat downstream (reviews of the retreat history in the Ross, Amundsen, and Weddell Seas are given in the following publications: Anderson et al. (2014); Spector et al. (2017); Larter et al. (2014); Hillenbrand et al. (2014); Nichols et al. (2019); Johnson et al. (2019)).

## 5.2 Evaluation of ice-sheet models

Our data provide an opportunity to evaluate the performance of Antarctic ice sheet models in the WAIS interior, where there are few other constraints on past ice thickness. We compare our results from the Pirrit Hills and the Whitmore Mountains to five thermomechanical ice-sheet models as well as the ICE-6G_C reconstruction of ice-sheet history. Two of the thermomechanical models (Kingslake et al., 2018) are identical except that they have very different accumulation-rate histories, and therefore produce different ice-thickness histories in West Antarctica. The first, which Kingslake et al. (2018) refer to as the "reference simulation", is forced by an accumulation-rate history that is on average much higher over the past 35 kyr than what has been reconstructed from the WAIS Divide ice core. The second is forced by the WAIS Divide accumulation-rate record. The third model is by Pollard et al. (2017) and is the best-scoring member of a large ensemble of simulations that are scored by comparison to geological and modern observations. At the Pirrit Hills and Whitmore Mountains, this simulation is relatively similar to other simulations using the same model (Pollard et al., 2016; Pollard et al., 2018), which, for simplicity, are not shown here. The fourth thermomechanical model is by Tigchelaar et al. (2018) and is a 800 kyr simulation (we show the last 35 kyr) that is driven by a coarse-resolution three-dimensional climate model. The fifth model is from Whitehouse et al. (2012) and is a semi-transient simulation constrained by geological and glaciological observations. The last model is the ICE-6G_C reconstruction of ice-sheet history (Argus et al., 2014; Peltier et al., 2015), the Antarctic component of which is calibrated with a similar set of observations as used by Pollard et al. (2017) and Whitehouse et al. (2012).

Figure 7 shows the ice-thickness histories extracted from these models at the Pirrit Hills and Whitmore Mountains along with our constraints from these sites. We compare ice-thickness relative to present rather than absolute ice thickness or elevation relative to sea level because we are more interested in whether the models correctly simulate thickness changes during the last deglaciation and less interested in whether the present-day ice sheet is correctly represented.

At the Pirrit Hills, the best performing model is that of Pollard et al. (2017). The highstand in the model matches the depositional limit, and the subsequent thinning occurs only slightly earlier (within ~2 kyr) than indicated by the exposure-age constraints from Mt. Tidd (Fig. 7a). All of the other models depict highstands that are at least ~100 m too thick or too thin and thinning that occurs several millennia too early. Because thinning at the Pirrit Hills is expected to have been primarily paced by the retreat of grounded ice in the southern Weddell Sea, this suggests that the grounding line retreats too early in all of the thermomechanical ice-sheet model simulations. We note that premature thinning in the models could also be caused by underestimating the magnitude and/or rapidity of the deglacial rise in snowfall, which as discussed by Hein et al. (2016) may have delayed the onset of thinning.

At the Whitmore Mountains, two of the simulations are ruled out because they depict ice considerably more than 190 m thicker than present, which is the upper limit on the highstand imposed by the [14]C-saturated bedrock samples (Fig. 7b). To evaluate the timing of thickening and thinning, we use not only the [14]C constraints from the lowest two bedrock samples but also those from the sample ~130 m above the present-day ice surface (Figs. 7c, d). In comparison to the lower samples, the ~130 m sample provides slightly more restrictive constraints on ice cover and exposure. The two best-performing models at the Whitmore Mountains are (i) the simulation by Kingslake et al. (2018) that is forced by the WAIS Divide accumulation-rate

record, and (ii) the simulation by Pollard et al. (2017). Neither simulates ice-cover of the ~130 m sample, so the constraints in Figure 7d are not applicable to these models. The former model is the only simulation with sufficiently brief thicker-than-present ice cover to be permitted by the $^{14}$C concentrations of the two lowest samples from Mt. Seelig. However, it depicts thinning to the modern ice level prior to 7 kyr B.P., which, as discussed above, is earlier than when sites downstream of the Whitmore Mountains reached their modern ice levels. The latter model simulates the onset of ice-cover occurring a few millenia too early and/or re-exposure occurring a few millenia too late; but it successfully simulates thinning to the modern ice level within the past ~7 kyr. The other four models do not simulate burial of the two lowest samples by the WAIS during the past 35 kyr and therefore do not appear in Figure 7c. The Kingslake et al. (2018) reference simulation and the simulation by Tigchelaar et al. (2018) are consistent (or nearly consistent) with the $^{14}$C constraints of the ~130 m sample (Fig. 7d); however, these models, along with ICE-6G_C, do not capture the general timing and magnitude of thickness changes at the Whitmore Mountains. The model of Whitehouse et al. (2012) is consistent with the highstand constraints from the Whitmore Mountains, however, because the simulation only spans the past 20 kyr, we have limited ability to evaluate the timing of thickness changes in the model.

The overall best performing model at both the Pirrit Hills and the Whitmore Mountains is that of Pollard et al. (2017). This is perhaps not surprising because this is the best-scoring run of an ensemble of ice-sheet simulations that were scored based on how well they agreed with geological observations from many sites in Antarctica (though few from the ice-sheet interior). We note that the accumulation-rate history of this model at the WAIS Divide ice-core site is lower than the ice-core-derived accumulation-rate record (Fudge et al., 2016). Therefore, the performance of the model, especially near the divide, could probably be improved with a more realistic forcing.

## 6 Conclusions

We present cosmogenic-nuclide constraints on ice thickness changes since the LGM from the Pirrit Hills and Whitmore Mountains, located on the flank and the divide of the WAIS, respectively. At the Pirrit Hills, monotonic thinning occurred after accumulation rates had risen from their ice-age low, implying that the dominant control on ice thickness was the retreat of the ice-sheet margin downstream. In contrast, at the Whitmore Mountains, the WAIS appears to have initially thickened following the LGM due to the increased snowfall, and only thinned once the dynamic effects of margin retreat began to outpace the thickening from snowfall. We compare our ice-thickness constraints to several recently-published models of the Antarctic ice sheet over the last deglaciation and find that while most of the models poorly capture the timing and/or magnitude of thickness changes at the Pirrit Hills and Whitmore Mountains, the model of Pollard et al. (2017) performs well at these sites, which, in part, is likely due to the fact that it is calibrated with geological observations of ice thickness change.

*Data availability.* Sample information and cosmogenic-nuclide data are available in the ICE-D: ANTARCTICA database (http://antarctica.ice-d.org)

*Author contributions.* PS and JS conducted the fieldwork. BG made the carbon-14 measurements. PS made the beryllium-10 measurements, analyzed all data, and wrote the manuscript.

*Competing interests.* The authors declare that they have no conflict of interest.

*Acknowledgements.* Support for this work was provided by U.S. National Science Foundation (NSF) grants 1142162 and 1341728 and the
5    United States Antarctic Program. P.S. received support from the NSF Graduate Research Fellowship Program. We thank Trevor Hillebrand, Mika Usher, Taryn Black, and Maurice Conway for assistance in the field; Kier Nichols for lab assistance; Greg Balco and Eric Steig for insightful discussions; and David Pollard, Torsten Albrecht, Jonathan Kingslake, and Michelle Tigchelaar for providing ice-sheet simulations. Geospatial support for this work provided by the Polar Geospatial Center under NSF-OPP awards 1043681 and 1559691.

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

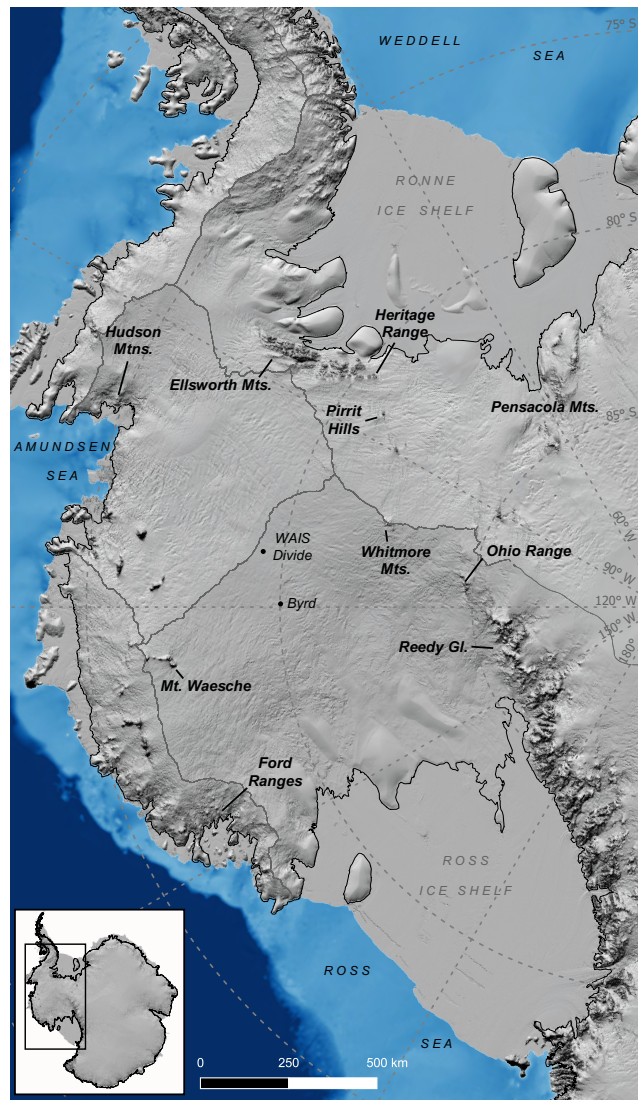

**Figure 1.** Map of West Antarctica. Hillshade of ice-sheet surface topography (Howat et al., 2019) is vertically exaggerated 20× and is overlaid on marine bathymetry (Fretwell et al., 2013). The continental shelf is shown in light blue. The grounding line (Bindschadler et al., 2011) and major ice divides (Zwally et al., 2012) are traced in gray.

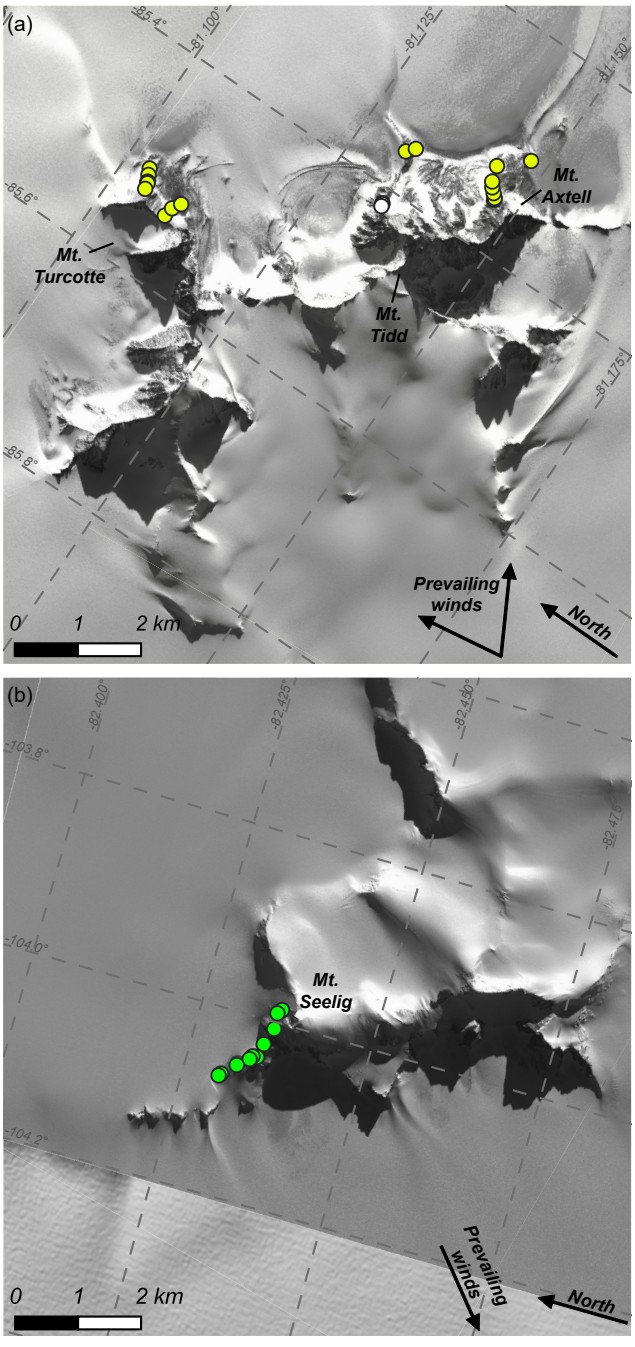

**Figure 2.** WorldView satellite imagery (copyright 2012 DigitalGlobe, Inc.) of the Pirrit Hills **(a)** and Mt. Seelig in the Whitmore Mountains **(b)**. Circles show the locations of samples discussed in the text, and their colors correspond to the circles in Figure 3. Wind direction arrows are based on the orientation of snowtails visible in the satellite imagery. The range of wind directions shown in panel **(a)** reflects the fact that the wind orientation varies with location around the mountains. The wind direction is relatively constant in panel **(b)** and so a single vector is used.

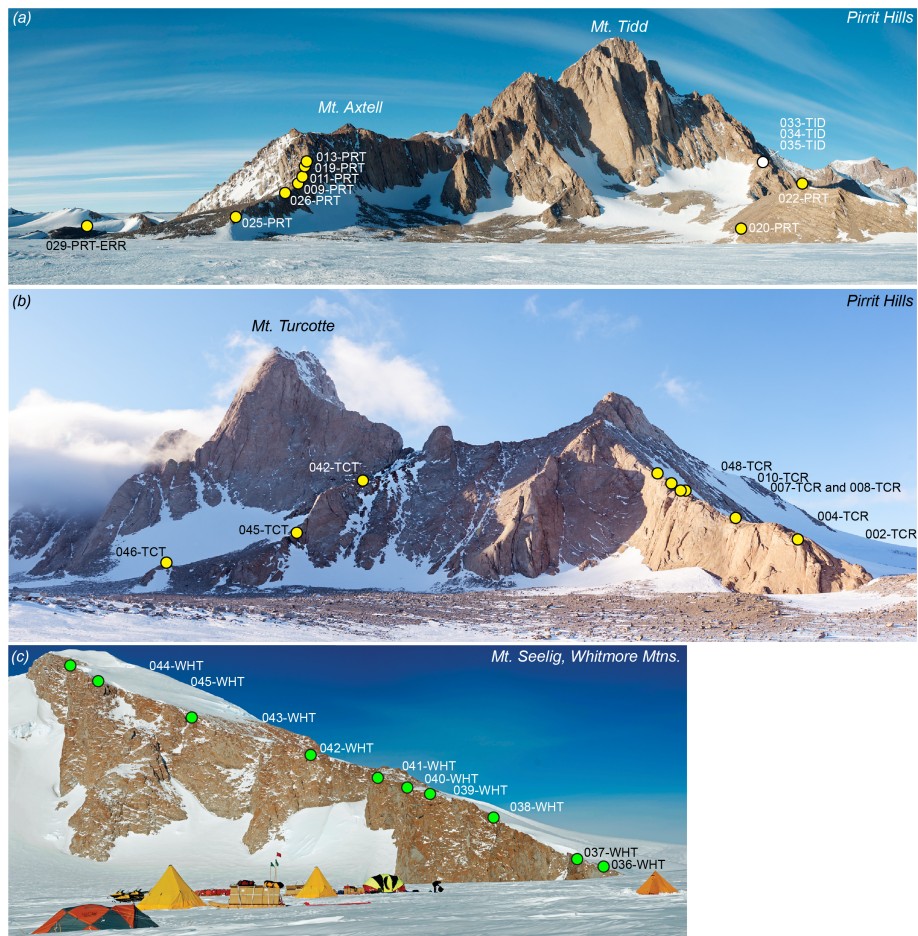

**Figure 3.** Views facing south of Mts. Axtell and Tidd **(a)** and facing west of Mt. Turcotte **(b)** at the Pirrit Hills. **(c)** View to the south of the northwest ridge of Mt. Seelig. Note that all circles in **(a)** and **(b)** denote samples of glacial deposits, while circles in **(c)** denote bedrock samples. The white circle in **(a)** represents samples from the depositional limit on Mt. Tidd, which have not been analyzed. Circles are labeled with abbreviated sample names. Mt. Turcotte samples, as well as the unanalyzed samples from Mt. Tidd, begin with the prefix "16-PRT-" (e.g. 16-PRT-042-TCT); all other samples begin with the prefix "13-NTK-".

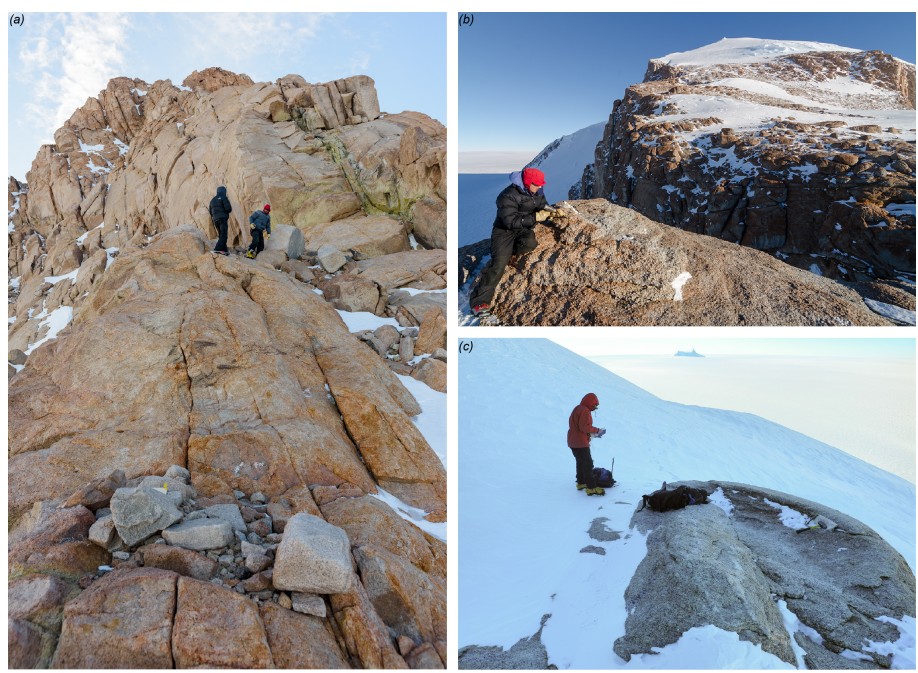

**Figure 4. (a)** View looking up the steep NE buttress of Mt. Axtell at the Pirrit Hills. The granite ridge crest is generally a few meters wide, oxidized, and, in places, displays evidence of exfoliation. In contrast, the glacial deposits, which are visible in the foreground as well as next to the two geologists, are typically only lightly weathered. The boulder with a very flat upper surface in the central foreground is sample 13-NTK-019-PRT. The depositional limit, where sample 13-NTK-013-PRT was collected, is ~15 m higher, near the level of the two geologists. **(b)** View looking up the NW ridge of Mt. Seelig in the Whitmore Mountains. The geologist is collecting sample 13-NTK-041-WHT (236 m above the modern ice surface) from the bedrock knob. Other samples come from the narrow ice-free strip of bedrock close to the cliff edge that is visible in the background. **(c)** Photo of bedrock sample 13-NTK-044-WHT, the highest elevation sample from Mt. Seelig. The sample was collected from an outcrop only a few meters wide that is likely kept ice-free by strong wind near the cliff edge.

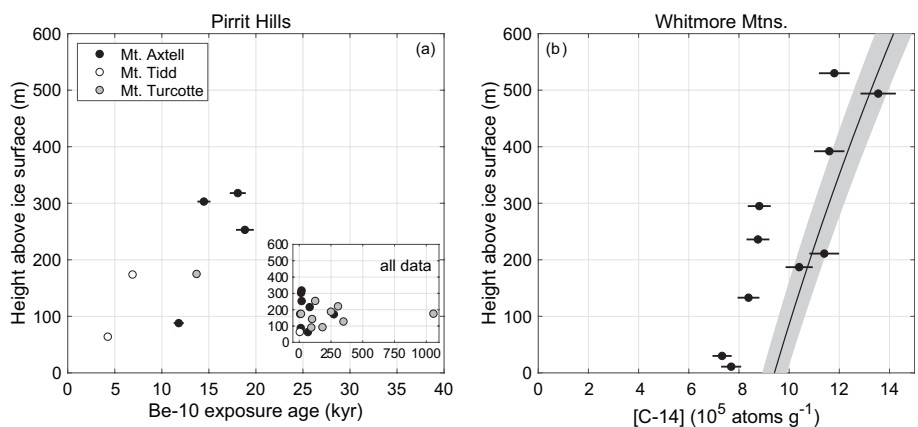

**Figure 5.** **(a)** Beryllium-10 exposure age of erratics from the Pirrit Hills plotted against their height above the modern ice surface. Inset shows the apparent ages of all glacial deposits, the majority of which are pre-exposed. **(b)** Carbon-14 concentrations in bedrock samples from the Whitmore Mountains plotted against their height above the modern ice surface. The black line and surrounding gray band represent calculated $^{14}$C saturation concentrations, which are a function of elevation, and their uncertainty. As discussed in the text, $^{14}$C saturation occurs after ~30-35 kyr of continuous exposure, at which point nuclide production is balanced by decay, and the $^{14}$C concentration is no longer time dependent. Error bars for both **(a)** and **(b)** are one standard error.

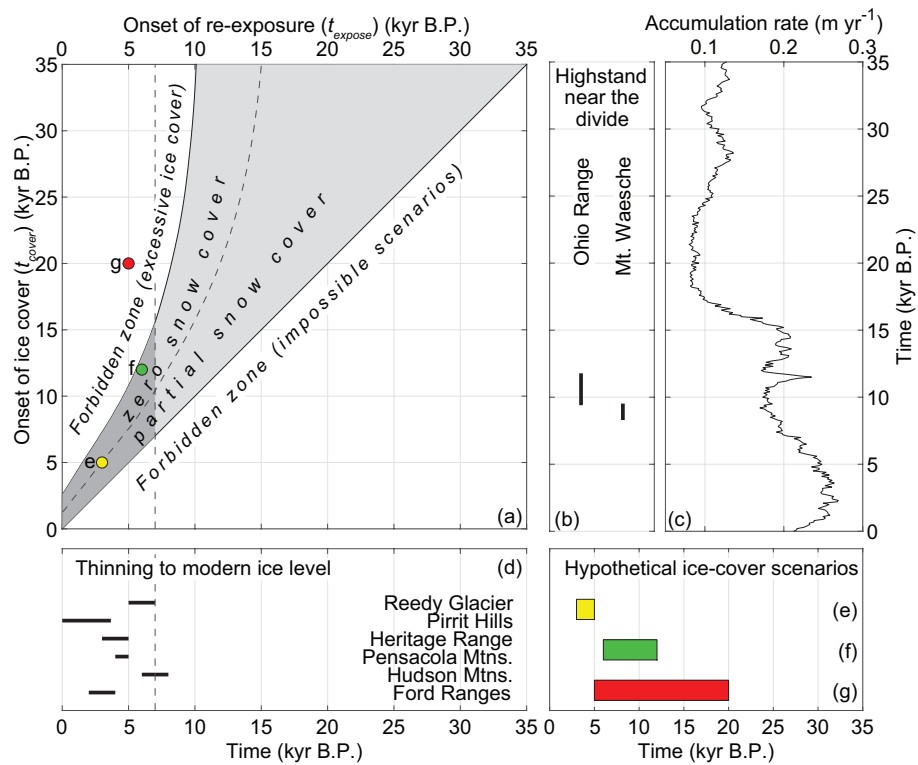

**Figure 6. (a)** Chronological constraints on exposure and ice cover of the two bedrock samples within ∼30 m of the modern ice surface at Mt. Seelig. Scenarios that plot in the gray regions are permitted by the [14]C concentrations. The lower-right half of the diagram is forbidden because re-exposure cannot occur prior to ice cover; the white area on the left side of the diagram is forbidden because these scenarios underpredict the observed [14]C concentrations. The gray areas left of the curved dashed line assume that less-than-saturated [14]C concentrations are only due to cover by WAIS thickening, while the areas to the right allow for the possibility that the samples experienced cover by a combination of a thicker WAIS and expanded snow fields. The position of these regions accounts for $1\sigma$ measurement uncertainties of both samples. The 1:1 line represents end-member scenarios of zero ice-sheet cover, which are permitted by the [14]C data. The three circles correspond to the hypothetical ice-cover scenarios shown in panels **(e-g)**. **(b)** Timeline showing when ice was at a highstand at the Ohio Range and at Mt. Waesche, sites near the WAIS divide, based on exposure dating of glacial deposits (Ackert et al., 1999, 2007, 2013). Note that the highstands could have begun before and/or persisted after the ages shown. **(c)** The accumulation-rate record from the WAIS Divide ice core (Fudge et al., 2016). **(d)** Timeline showing when modern ice levels were established at nunatak sites in West Antarctica. The vertical dashed lines in panels **(a, d)** represent the earliest time (7 kyr B.P.) that the modern ice level could have been reached at the Whitmore Mountains, which is based on constraints from Reedy Glacier, the Pirrit Hills, and the Heritage Range, sites that share similar flow paths to the Whitmore Mountains. The dark gray region in panel **(a)** represents scenarios that are both (i) permitted by the [14]C concentrations, and (ii) consistent with the constraints shown in panel **(d)**. **(e-g)** Timelines showing hypothetical burial intervals of the lowest two Mt. Seelig samples, which correspond to the circles in panel **(a)**.

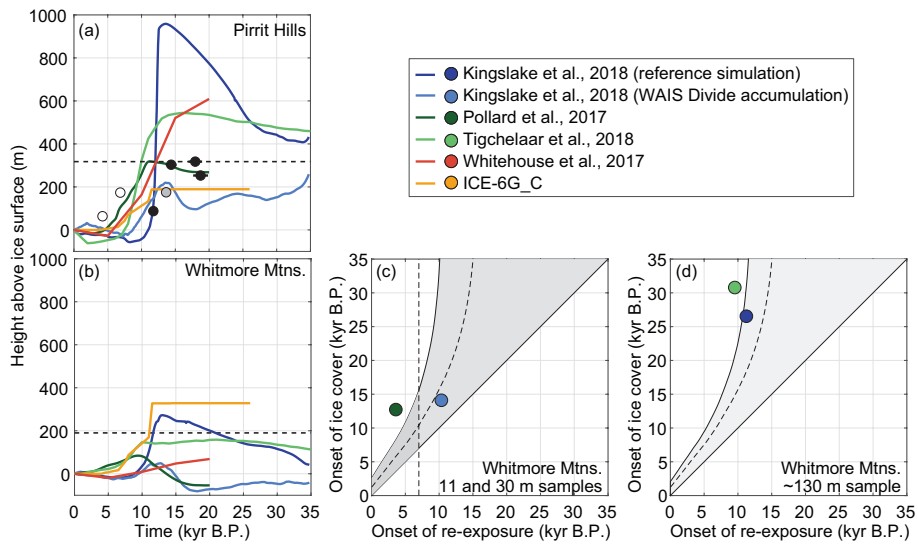

**Figure 7.** Evaluation of ice-sheet models at the Pirrit Hills and Whitmore Mountains. Panels **(a)** and **(b)** show ice-thickness histories from five ice-sheet models at the Pirrit Hills and Whitmore Mountains, respectively. The horizontal dashed line in panel **(a)** represents the height of the highstand, while in panel **(b)**, it represents the upper limit of the highstand. **(a)** also depicts [10]Be exposure ages as in Figure 5a. **(c)** shows chronological constraints on exposure and ice cover from the two lowest elevation bedrock samples. The gray areas are permitted by the bedrock [14]C data, and the vertical dashed line represents the earliest time (7 kyr B.P.) that the modern ice level could have been reached at the Whitmore Mountains. Refer to the caption of Figure 6 for a full explanation of this figure. **(d)** shows the same constraints, except from the bedrock sample ∼130 m above the ice surface. In comparison to the lower samples, the ∼130 m sample provides slightly more restrictive constraints on ice cover and exposure. For panels **(c)** and **(d)**, only models that simulate both cover and re-exposure of the sample(s) within the past 35 kyr appear on the diagrams.