# Peer review of "Thickness of the divide and flank of the West Antarctic Ice Sheet through the last deglaciation"

_The Cryosphere, 2019_

## Referee Comment (RC1) · Anonymous Referee #1 · 3 Aug 2019

Spector et al. report new data from the Weddell Sea sector and from the Ross Sea-Weddell Sea ice divide in West Antarctica. The last glacial maximum (LGM) highstand is determined from the elevation of young (post-60 kyr BP) Be-10 and saturated C-14 nuclide concentrations, and modelled scenarios of past ice thickness change at the sites. The authors then interpret these respective ice thickness changes to record a combination of snowfall accumulation and dynamic thinning through the deglaciation.

The paper is generally well-written with clear figures and adequate provision of supplementary data. The data are new with robust lab, measurement and calculation procedures, and the approach to determine ice thickness change from these data is novel. The data are worthy of publication and the conclusions would be of interest to the readers of The Cryosphere. However, I have some concerns that need to be addressed,

which have potential implications for the reliability of the paper's conclusions.

Main comments:

1. Treatment of Be-10 and C-14 data

As the authors point out, C-14 has an advantage over Be-10 owing to the shorter decay time, which prevents most inheritance from pre-exposure from being recorded in the sample concentrations. At Pirrit Hills, the Be-10 ages are used to indicate when the local ice surface reached a highstand (e.g. before 18 kyr BP; page 6, lines 19-21). Yet, it is noted for other studies that the use of C-14 has highlighted that approx. LGM Be-10 ages are not reliable indicators of the highstand (page 7, lines 3-5). The authors do not acknowledge the possibility that their Be-10 ages may similarly record minor inheritance, which would therefore make them minimum age estimates and less reliable highstand indicators.

Samples with C-14 concentrations that are indistinguishable from saturation are used to delimit the LGM highstand at Whitmore Mountains (e.g. page 7, lines 8-9). The Nichols et al. TCD paper (which shares authors with this paper) discusses how high C-14 concentrations can result from analytical and geomorphic issues – what is there to say that any of these saturated samples do not suffer from the same issues? Can we trust this approach for delimiting the LGM highstand?

The delimitation of the highstand has implications for the interpretation of ice thickness change (e.g. which modelled scenarios are ruled out).

2. Calculation of ice-cover and exposure history scenarios

There is little discussion about the production rate in the method (page 7, line 31 to page 8, line 24). Do these model scenarios account for sample-specific production rate differences? It appears that production is calculated for spallation only, without any mention of production from muons. Muonic production near the surface is important for C-14 (Lupker et al., 2015; Hippe, 2017) – is this accounted for in these calculations?

Minor comments:

Page 1, lines 10-11 (abstract): The final sentence would be better as a conclusion/implications. Perhaps something about how the model(s) calibrated with geological observations performed best.

Page 1, lines 22-23: Got a suitable reference for this sentence?

Page 2, lines 2-4: Is there evidence of atmospheric temperatures being warm enough to induce thinning in this region of Antarctica?

Page 3, lines 19-20: Or, similar to what you mention later, the abundance of debris may relate to debris source (i.e. the ice sheet had access to more debris when near to its highstand).

Page 3, line 33 to page 4, line 1: This is an important point, but the data to support it has not been published. This point should be removed or the data included in the Supplement with suitable description and analysis to reach this conclusion.

Page 4, lines 23-27: As above. Is it necessary to include the data if the data "are not relevant to this paper"?

Page 6, lines 6-7: Confusing sentence. Is CRONUS-A from the Whitmore Mountains or the McMurdo Dry Valleys?

Page 7, lines 11-14: Overly complex sentence which could be made simpler for the reader.

Page 7, lines 16-17: Can you provide a reference for the typical West Antarctic snow density?

Page 9, lines 13-15: It is dangerous to use ice sheet models to support interpretations of data, and then use that data to assess the quality of the models. Reconsider using models to interpret the data.

Page 9, line 35: "ice" typo.

Page 10, lines 9-10: Most of these cited studies do not record grounding-line changes.

Page 11, lines 16-17: Yes, perhaps this suggests that the grounding line retreated too early, but, as you say previously, thinning is also dependent on changes in the accumulation rate.

Page 12, lines 1-3 and lines 15-16: Why not compare to other models calibrated with geological observations (e.g. Whitehouse et al., 2012; Briggs et al., 2014)? This would strengthen the conclusion that the best performing models are calibrated with geological observations.

Figure 7: You should explain what the vertical dashed line represents in plot c.

[Figure]

---

## Referee Comment (RC2) · Anonymous Referee #2 · 12 Aug 2019

This well-written and well-considered manuscript contrasts LGM ice thickness changes from the centre of the West Antarctic Ice Sheet with those closer to its margins. The authors reveal new cosmogenic nuclide-derived constraints on ice sheet thickness at two unstudied locations (Pirrit Hills and Whitmore Mtns) derived through 10Be surface exposure dating and novel interpretations of in situ 14C data. The authors find support for a post-LGM highstand in the centre of the ice sheet as a likely consequence of increased precipitation during deglaciation, followed by dynamic thinning that propagated upstream from the ice sheet margin. The idea is not new, and there are other field studies from interior regions of the WAIS (Steig et al., 2001; Ackert et al., 1999), the Ross Sea (Todd et al., 2010; Hall et al., 2015) and Weddell Sea (Hein et al., 2016) and that find similar evidence for this balance between oceanic and atmospheric con-

trols on ice sheet behaviour during deglaciation. The manuscript adds further support for this idea, obtained through novel application and interpretation of in situ 14C from a rare, interior WAIS location. The modelling of in situ 14C exposure histories is novel and will be of wide interest to those who work with cosmogenic nuclide data. The conclusions are supported by the data, and should be of wide interest to the readers of the Cryosphere. I recommend publication after addressing some minor points.

Specific comments:

- The mix of saturated and finite 14C ages on Mt. Seelig is perplexing and requires further discussion. The saturated 14C ages are used to "eliminate" the possibility that the WAIS was thicker than 190m (elevation of the lowest saturated age) during the LGM, without discussion on the reliability of those ages. It is worth discussing the reliability of those saturated ages given the recent study by Nichols et al. (2019), who repeated measurements of 14C in samples that were previously reported as saturated (Balco et al., 2016) and got finite ages. As the data come from the same lab, further discussion is warranted.

The authors consider the proximity to local ice caps and snowfields to explain the finite 14C ages at higher altitudes, with distances of 5-20m reported. However, it is not discussed whether the saturated samples are located further away from the snowfields? Is there any sort of correlation here that could be used to support this argument?

- The authors cite unpublished Al/Be/Ne data to support their argument for low erosion rates and long exposure of bedrock surfaces in the Whitmore Mtns. The authors should either publish this data within this manuscript, or remove reference to them.

- Comment on Section 5.1: the argument for simple monotonic post-LGM thinning of ice sheet margins despite a deglacial increase in snowfall is likely correct, but is overly simplified. In the Heritage Range, the competition between increased snowfall and dynamic thinning is argued to explain the delayed deglaciation at that site, which initiated in earnest only after ∼10 ka when dynamic thinning began to outpace deglacial

increases in snowfall (Hein et al., 2016). If correct, the influence of increased snowfall is more widespread than implied (i.e., it's influence extends beyond the divide), even though it may not have caused local thickening. Similar competition between processes has been suggested to explain ice histories in the Ross Sea sector (Hall et al., 2015; Todd et al., 2010).

Technical corrections:

- What is the distance of Pirrit Hills from the grounding line and the divide?

-p3 line 30- statement that no glacially-transported cobbles found contrasts with next statement on discovery of indurated till. Rewrite to clarify.

- P4L8 – replace "remove" with "minimize"

-P4L15 – Perhaps include in Table the distance of each sample to the present snow/ice cap and reference here.

-P4L19 –For clarity, state from where the ice margin elevation was determined at each site (given this can vary significantly around a nunatak).

- P8L10 – can't use unpublished data to support your argument.

- P9L35 – spelling of "ice"

---

## Editor Decision (ED1)

I would like to thank both reviewers for their detailed and constructive comments on this manuscript and also the authors for posting their response to the reviewers' comments.

Several issues are not fully resolved by the authors' response to the reviewers' comments. I note these below, along with a few additional points that the authors may want to consider as they prepare a revised version of their manuscript.

Reviewer 1 raises some concerns that have potential implications for the reliability of the paper's conclusions. I share some of these concerns, and in particular I feel that you do not always provide a full discussion of the uncertainties on the data or explore alternative explanations for the evidence. Some of the assumptions you make when interpreting the data have significant implications for the exposure/burial history that you subsequently infer. I encourage you to acknowledge the limitations of the data more thoroughly and discuss the viability of alternative scenarios where relevant.

In general, both reviews are positive, and they highlight the novelty and importance of this study. I therefore encourage you to submit a revised manuscript that addresses the points mentioned below and in the individual reviews.

Kind regards,

Pippa Whitehouse

Specific points

- Both reviewers comment on the reliability of the saturated ages reported on page 7, lines 7-9. You provide an extensive discussion on this point in your response to the reviewers' comments, but you also state that you do not plan to include any additional information in the manuscript. Given that both reviewers commented on this, and given the importance of these ages in determining the ice history at Mt Seelig, I think it is important to include a brief discussion on the reliability of the saturated ages in the main text - if only for the benefit of those readers who are unfamiliar with the issues associated with analysing and interpreting such samples. As you note, the full details can remain in the review documents.

- In two places Reviewer 2 mentions a number of earlier articles that discuss the competing roles of precipitation change and grounding line dynamics in controlling post-glacial Antarctic change. Of the articles mentioned by the reviewer, all are already cited in the original version of your manuscript except Hall et al. (2015, Nature Geoscience). I encourage you to consider including a reference to this highly relevant article in the revised version of your manuscript.

- Reviewer 1 queried whether there is evidence of atmospheric temperatures being warm enough to induce thinning across West Antarctica. Much of your response (and the text in the manuscript) appears to rely on the assumption that there is a delay of 10-30 kyr between atmospheric warming and ice thinning. You include a reference to an entire textbook, which makes it difficult to determine the precise basis for this assumption, but the text on line 4 (page 2) suggests that it relates to the time required for surface warming to have an impact on conditions at the base of the ice sheet. However, on lines 2-3 (page 2) you also mention the process by which an increase in ice temperature (at any depth) will change the rheology of the ice, thus allowing it to deform and flow more easily. The time lag for this second process is presumably much shorter, perhaps negating your assumption that

there must be a delay of at least 10 kyr between warming and thinning? And in fact, I don't think the reviewer is even asking whether warming-induced thinning has commenced, but rather whether the deglacial increase in atmospheric temperature was sufficient to trigger thinning by one of the processes described above. Please address this second point.

- Opening sentence of section 5.1: "…despite the deglacial increase in snowfall…" It is not clear what evidence you are drawing on to support this statement, but elsewhere in the manuscript I note that you refer to the WAIS Divide ice core when discussing accumulation change across West Antarctica. The Pirrit Hills are in a different catchment to the WAIS Divide ice core (figure 1 of your manuscript) and hence they may have experienced a different snowfall history to that at WAIS Divide (page 10, line 24 of your manuscript). The statement at the start of section 5.1 therefore requires additional justification if you wish to use accumulation change at WAIS Divide as a proxy for accumulation change at the Pirrit Hills. If you are drawing on alternative evidence to support the statement about accumulation change at the Pirrit Hills then please make this clear. In light of my comments, please also check the robustness of the statement in the conclusions that refers to accumulation rates at the Pirrit Hills.

- On page 9, you draw on evidence from sites across West Antarctica to support your inference that ice was previously thicker in the Whitmore Mountains. Considering the likely flowlines of the ice sheet during the last deglacial period, it is not clear to me that ice thickness changes at Mt Waesche (page 9, line 16) should necessarily be similar to those at the Whitmore Mountains. Similarly, one could envisage a scenario whereby ice was thicker than present at Byrd Station during the LGM (page 9, line 19) but not at the divide upstream of this site. It would be useful if you could include a statement about the degree to which ice thickness changes at Mt Waesche and Byrd Station can be expected to co-vary with ice thickness at the Whitmore Mountains (as you do when discussing evidence from the Ohio Range).

- Page 9, line 33: "Thinning to the modern ice level at Mt. Seelig therefore could not have occurred before 7 kyr ago". To improve the clarity of your argument, please be more explicit about which of the constraints mentioned in the previous paragraph you are drawing on to make this quantified statement.

- Page 10, line 10: could the ice have been thicker than present for a brief period during the LGM? i.e. could it be that the samples were not completely saturated at the beginning of the ~15ka burial period?

- Page 10, line 31: ICE-6G is not really a 'model of glacial isostatic adjustment'; it is an ice history model in the sense you are using it

- Please include latitude and longitude labels on figure 1

---

## Editor Decision (ED2)

Editor comments

I would like to thank the authors for addressing all comments on the previous version of the manuscript. In particular, the interpretation of your results is much clearer and your overall conclusions are more robustly justified.

In light of your edits, there are two points that require additional clarification:

1) Page 10, lines 26-27: you mention that your results are based on the assumption that the samples close to the present ice surface have "experienced only one period of exposure during the past ~30 kyr". This implies that you assume the samples were previously covered, but the timing or duration of this coverage is not clear. Please address this. In addition, in light of the text on lines 24-25 please make it clear which "pair of samples" you are referring to on line 26, and if appropriate adopt the terminology introduced above that refers to a "simple exposure age".

2) Page 12, lines 34-35: In the previous paragraph you mention that it is possible the samples were not 14C-saturated when they were last covered, i.e. they could have been briefly covered during the LGM, then uncovered, and covered again at 15 ka BP. Given this possibility, please re-consider the robustness of your statement that the "constraints demonstrate that the WAIS at the Whitmore Mountains was the same thickness or thinner than present during the LGM". Please also clarify what time period you are referring to when you talk about the LGM (apologies if I have missed this).

I list below a number of minor technical points that also require attention. Once these and the issues above are addressed, I would be happy to review a revised version of the manuscript.

Kind regards,

Pippa Whitehouse

Minor technical points (suggested edits in **bold**)

- Page 2, line 29: delete 'also' as there is no previous mention that data are scarce
- Page 3, line 17: 'isolated **glacial** deposits'
- Page 6, line 12-13: 'of **the** UW standards'
- Page 8, line 13: 'was **at** least'
- page 10, line 25: revise back to '130 m **above the ice surface**'
- Page 10, line 31: delete one instance of 'our' and edit 'initial' -> 'initially'
- Page 11, line 19: clarify what type of deformation you are referring to, presumably ice
- Page 11, line 22: the reference to Hall et al. (2015) seems a little out of place in this sentence that specifically talks about the Heritage Range and Pirrit Hills. It may sit more appropriately with the other references listed on page 2, line 19
- Page 12, line 28/fig. 6b: in line with an earlier edit, remove references to Mt. Waesche
- Figure 2: refer to (a) and (b) in the first sentence of the caption
- Figure 7: add labels (a) to (d) on appropriate panels

---

## Author Response (AR3)

**Author responses to referee comments on "Thickness of the divide and flank of the West Antarctic Ice Sheet through the last deglaciation"**

Perry Spector et al.
Pspector@bgc.org

Referee comments and our responses are shown in black and blue, respectively.
* * *
**Response to Anonymous Referee #1**

Spector et al. report new data from the Weddell Sea sector and from the Ross SeaWeddell Sea ice divide in West Antarctica. The last glacial maximum (LGM) highstand is determined from the elevation of young (post-60 kyr BP) Be-10 and saturated C-14 nuclide concentrations, and modelled scenarios of past ice thickness change at the sites. The authors then interpret these respective ice thickness changes to record a combination of snowfall accumulation and dynamic thinning through the deglaciation. The paper is generally well-written with clear figures and adequate provision of supplementary data. The data are new with robust lab, measurement and calculation procedures, and the approach to determine ice thickness change from these data is novel. The data are worthy of publication and the conclusions would be of interest to the readers of The Cryosphere. However, I have some concerns that need to be addressed, which have potential implications for the reliability of the paper's conclusions.

*Thank you for the careful review of the manuscript and helpful comments. We have incorporated them as described below.*

Main comments:

1. Treatment of Be-10 and C-14 data

As the authors point out, C-14 has an advantage over Be-10 owing to the shorter decay time, which prevents most inheritance from pre-exposure from being recorded in the sample concentrations. At Pirrit Hills, the Be-10 ages are used to indicate when the local ice surface reached a highstand (e.g. before 18 kyr BP; page 6, lines 19- 21). Yet, it is noted for other studies that the use of C-14 has highlighted that approx. LGM Be-10 ages are not reliable indicators of the highstand (page 7, lines 3-5). The authors do not acknowledge the possibility that their Be-10 ages may similarly record minor inheritance, which would therefore make them minimum age estimates and less reliable highstand indicators.

*To address this, we have added the following sentences to the end of the first paragraph of Section 4.1.*
> *"Strictly, we cannot rule out the possibility that the youngest ages also record minor prior exposure. However, the similarity of these ages to the youngest ages from the nearby*

*Heritage Range (discussed below) suggests that this effect is either absent or minor, amounting to less than 1-2 kyr."*

Samples with C-14 concentrations that are indistinguishable from saturation are used to delimit the LGM highstand at Whitmore Mountains (e.g. page 7, lines 8-9). The Nichols et al. TCD paper (which shares authors with this paper) discusses how high C-14 concentrations can result from analytical and geomorphic issues – what is there to say that any of these saturated samples do not suffer from the same issues? Can we trust this approach for delimiting the LGM highstand?

The delimitation of the highstand has implications for the interpretation of ice thickness change (e.g. which modelled scenarios are ruled out).

*This comment is similar to a comment by Referee #2. Both comments will be addressed below in the response to Referee #2.*

2. Calculation of ice-cover and exposure history scenarios

There is little discussion about the production rate in the method (page 7, line 31 to page 8, line 24). Do these model scenarios account for sample-specific production rate differences? It appears that production is calculated for spallation only, without any mention of production from muons. Muonic production near the surface is important for C-14 (Lupker et al., 2015; Hippe, 2017) – is this accounted for in these calculations?

*This is a good point. Yes, the calculation does include $^{14}C$ production by spallation as well as from muon interactions. During times when the samples are simulated as being buried by ice or firn, we assume that all production ceases. This is obviously not fully realistic. However, because the thickness of overlying ice or firn is unknown, it is not possible to make a realistic estimate of the $^{14}C$ production rate by muons during times of burial. The effect of this simplification is minor. As discussed in Section 4.2, cover by only a few meters of firn is sufficient to block the majority of the cosmic-ray flux. We have changed the text to indicate that the production rate, P, includes both spallation and muon interactions.*

Minor comments:

Page 1, lines 10-11 (abstract): The final sentence would be better as a conclusion/implications. Perhaps something about how the model(s) calibrated with geological observations performed best.

*This is also a good point. We have added the following two sentences:*
> *"Most of the models we consider do not match the observed timing and/or magnitude of thickness change at these sites. However, one model performs well at both sites, which*

*may, in part, be due to the fact that it was calibrated with geological observations of ice thickness change from other sites in Antarctica."*

Page 1, lines 22-23: Got a suitable reference for this sentence?

*We have added a citation to Cuffey & Patterson (2010).*

Page 2, lines 2-4: Is there evidence of atmospheric temperatures being warm enough to induce thinning in this region of Antarctica?

*Surface temperature, as reconstructed at the WAIS Divide ice-core site, increased primarily between ~20 and ~15 kyr B.P. (Cuffey et al., 2016). Therefore, thinning due to warming may have either (i) not yet commenced to a significant degree or (ii) begun sometime during the Holocene. The only sites we are aware of that provide information about this question are the nunataks near the margin of lower Reedy Glacier in the southern Transantarctic Mountains. As summarized in Section 5.1 of the manuscript,*

> *"Exposure dating at these sites establishes that thinning coincided with deglaciation of a large portion of the Ross Sea 9-7 kyr B.P (Spector et al., 2017; Todd et al., 2010). By ~7-6 kyr B.P., most of the thinning was complete; the ice sheet stood within ~50 of the present-day surface, down from a highstand that was at least ~150 m above present and likely 200-250 m or higher based on the height of depositional limits farther upstream on Reedy Glacier (Todd et al., 2010)."*

*The scenario in which thinning due to climatic warming began during the Holocene suggests that subsequent surface lowering would have occurred at these sites following ~7 kyr B.P. Such lowering is not observed. Exposure dating of erratics collected very near to the present-day ice level at these sites demonstrate that the ice surface has remained stable since ~7 kyr B.P. (Spector et al., 2017; Todd et al., 2010), which is consistent with the idea that thinning due to warming of the surface has not yet commenced in West Antarctica.*

*To clarify the manuscript, we have added the following sentence to the third paragraph of Section 1:*
> *"Exposure ages from small nunataks on lower Reedy Glacier demonstrate that ice levels have remained stable since ~7 kyr B.P. (Spector et al., 2017; Todd et al., 2010), suggesting that thinning due to climatic warming following the LGM has not yet commenced to a significant degree in West Antarctica."*

Page 3, lines 19-20: Or, similar to what you mention later, the abundance of debris may relate to debris source (i.e. the ice sheet had access to more debris when near to its highstand).

*Variations in debris abundance at different heights could, in part, be due to differences in debris source, as suggested by the reviewer; however, we think the simplest explanation is that debris*

*abundance varies as a function of the thinning rate and thus the amount of time the ice surface spends at a given height. At the Pirrit Hills, ice-sheet models predict a period of at least a few millennia of limited ice-thickness change when the WAIS is close to its highstand there, followed by more rapid thinning during the deglaciation (Figure 7). Given that thinning rates are generally expected to vary in this way, while there is no a priori expectation regarding the debris source, the most parsimonious explanation is that the abundance of debris is a function of the thinning rate.*

*This is essentially analogous to the idea that moraines and grounding-zone wedges accumulate when a glacier snout or ice-sheet margin resides in one position for a period of time. The absence of such features up-valley or shoreward are typically interpreted as an indication of the retreat rate, rather than as evidence that the glacier or ice-sheet ran out of rocks to deposit.*

*In this case, we have opted to not change the manuscript text.*

Page 3, line 33 to page 4, line 1: This is an important point, but the data to support it has not been published. This point should be removed or the data included in the Supplement with suitable description and analysis to reach this conclusion.

*The sentence about measurements of long-lived cosmogenic-nuclides on the till has been removed. We have replaced it with the following sentence: "The simplest interpretation is that the patch of weathered till is a remnant of a more extensive ancient deposit that has been largely eroded away."*

Page 4, lines 23-27: As above. Is it necessary to include the data if the data "are not relevant to this paper"?

*The reference to unpublished data has been removed.*

Page 6, lines 6-7: Confusing sentence. Is CRONUS-A from the Whitmore Mountains or the McMurdo Dry Valleys?

*We have changed this sentence to read: "CRONUS-A was collected from a slowly-eroding site in the McMurdo Dry Valleys (elevation: 1679 m; distance from Whitmore Mountains: 1650 km) that remained ice free during the LGM."*

Page 7, lines 11-14: Overly complex sentence which could be made simpler for the reader.

*We appreciate the suggestion, but we think that this sentence is relatively straightforward and works well in the context of the preceding and subsequent sentences.*

*Note that, in this sentence, we replaced "Because we can rule out that …" with "Because it is very unlikely that…" This was due to the fact that the discussion of unpublished cosmogenic-nuclide data was removed from earlier sections of the manuscript.*

Page 7, lines 16-17: Can you provide a reference for the typical West Antarctic snow density?

*We have added a reference to Mayewski et al. (2005), who describe the shallow cores recovered as part of the ITASE project.*

Page 9, lines 13-15: It is dangerous to use ice sheet models to support interpretations of data, and then use that data to assess the quality of the models. Reconsider using models to interpret the data.

*This is a good recommendation. We have removed the references to ice-sheet models in Section 5.1.*

Page 9, line 35: "ice" typo.
*Fixed*

Page 10, lines 9-10: Most of these cited studies do not record grounding-line changes.

*These studies record ice thickness changes upstream of the grounding line that were interpreted by the original authors to be related to grounding-line changes. We have added the words "appear to have" to the sentence in question. It now reads "...while grounding-line changes in the Weddell Sea sector appear to have continued into the late Holocene…"*

Page 11, lines 16-17: Yes, perhaps this suggests that the grounding line retreated too early, but, as you say previously, thinning is also dependent on changes in the accumulation rate.

*This is a good point. We have changed the end of this paragraph to read:*
>     *"Because thinning at the Pirrit Hills is expected to have been primarily paced by the retreat of grounded ice in the southern Weddell Sea, this suggests that the grounding line retreats too early in all of the thermomechanical ice-sheet model simulations. Premature thinning in the models could also be caused by underestimating the magnitude and/or rapidity of the deglacial rise in snowfall, which as discussed by Hein et al. (2016) may have delayed the onset of thinning."*

Page 12, lines 1-3 and lines 15-16: Why not compare to other models calibrated with geological observations (e.g. Whitehouse et al., 2012; Briggs et al., 2014)? This would strengthen the conclusion that the best performing models are calibrated with geological observations.

*This is also a good point. We have obtained model output for Whitehouse et al. (2012) for the Pirrit Hills and the Whitmore Mountains, and we have included this model in the evaluation.*

*Regarding the model of Briggs et al. (2014), we were only able to obtain model output of ice-surface elevation, not ice thickness, which is the variable we use in the analysis.*

Figure 7: You should explain what the vertical dashed line represents in plot c.

*The figure caption now explains what the vertical dashed line represents, along with the gray areas.*
* * *
**Response to Anonymous Referee #2**

This well-written and well-considered manuscript contrasts LGM ice thickness changes from the centre of the West Antarctic Ice Sheet with those closer to its margins. The authors reveal new cosmogenic nuclide-derived constraints on ice sheet thickness at two unstudied locations (Pirrit Hills and Whitmore Mtns) derived through 10Be surface exposure dating and novel interpretations of in situ 14C data. The authors find support for a post-LGM highstand in the centre of the ice sheet as a likely consequence of increased precipitation during deglaciation, followed by dynamic thinning that propagated upstream from the ice sheet margin. The idea is not new, and there are other field studies from interior regions of the WAIS (Steig et al., 2001; Ackert et al., 1999), the Ross Sea (Todd et al., 2010; Hall et al., 2015) and Weddell Sea (Hein et al., 2016) and that find similar evidence for this balance between oceanic and atmospheric controls on ice sheet behaviour during deglaciation. The manuscript adds further support for this idea, obtained through novel application and interpretation of in situ 14C from a rare, interior WAIS location. The modelling of in situ 14C exposure histories is novel and will be of wide interest to those who work with cosmogenic nuclide data. The conclusions are supported by the data, and should be of wide interest to the readers of the Cryosphere. I recommend publication after addressing some minor points.

*Thank you for the thorough review and constructive comments. We have addressed each comment as described below.*

Specific comments:

- The mix of saturated and finite 14C ages on Mt. Seelig is perplexing and requires further discussion. The saturated 14C ages are used to "eliminate" the possibility that the WAIS was thicker than 190m (elevation of the lowest saturated age) during the LGM, without discussion on the reliability of those ages. It is worth discussing the reliability of those saturated ages given the recent study by Nichols et al. (2019), who repeated measurements of 14C in samples that were previously reported as saturated (Balco et al., 2016) and got finite ages. As the data come from the same lab, further discussion is warranted.

*As the reviewer mentions, some replicate $^{14}C$ measurements on Antarctic samples prepared at Tulane University scatter significantly more than expected from their analytical uncertainty alone. This problem, however, does not appear to plague samples at random, but rather it has only been observed on glacial erratics collected from the Schmidt Hills (Nichols et al., 2019; see figure below) and on samples from Hatherton/Darwin Glaciers (unpublished; not shown in figure). Other replicate analyses on Antarctic samples prepared at Tulane University do not exhibit this problem (see figure). For example, $^{14}C$ concentrations measured on the Antarctic rock standard, CRONUS-A, scatter by 5.2% (n=13; not shown in figure; Goehring et al., 2019).*

[Figure]

*Figure 1. $^{14}C$ concentration plotted against sample elevation for samples prepared at Tulane U. on which replicates have been measured.The black line shows predicted $^{14}C$ saturation concentrations, which are a function of elevation. White circles represent glacial erratics from the Schmidt Hills; gray circles represent bedrock samples from various sites in Antarctica.*

*Because of the possibility of contamination from modern carbon, it is more likely to measure an erroneously high $^{14}C$ concentration than an erroneously low concentration. Experiments with different sample preparation procedures by Nichols & Goehring (2019) suggest that, in some cases, excess scatter is due to contamination by modern carbon. The excess scatter exhibited by replicate analyses on samples from Hatherton/Darwin Glacier (mentioned above) is probably due to insufficient removal of organic surfactants used for quartz purification (Nichols & Goehring, 2019). Although samples from the Whitmore Mountains were exposed to these same surfactants, Nichols & Goehring (2019) have shown that the acid-etching procedure we followed sufficiently removes organic contaminants. Although samples from the Schmidt Hills (figure) were not exposed to surfactants, the replicates that produced the lowest $^{14}C$ concentrations were measured on the most aggressively-etched quartz (similar to the treatment received by the Whitmore Mtns. samples). In some cases, contamination by modern carbon is expected to produce $^{14}C$ concentrations higher than predicted saturation concentrations. This is not observed for samples from the Whitmore Mountains. Rather, three samples spanning ~300 m of elevation have concentrations indistinguishable from saturation. To summarize, while we cannot*

*prove that the $^{14}C$ measurements from the Whitmore Mountains are accurate, for the reasons discussed above, we have no reason to believe that there is unrecognized analytical error.*

*We feel that discussion of this issue is beyond the scope of the manuscript, and we rely on the open discussion format of this journal to disseminate this information.*

The authors consider the proximity to local ice caps and snowfields to explain the finite 14C ages at higher altitudes, with distances of 5-20m reported. However, it is not discussed whether the saturated samples are located further away from the snowfields? Is there any sort of correlation here that could be used to support this argument?

*We did not measure the distance from samples to the nearest snowfields while in the field, but from our photographs and notes there is no obvious relationship between snowfield proximity and whether a sample is $^{14}C$ saturated or not. To clarify the manuscript, we have added the following paragraph to the end of Section 4.2.*

> *"While snow cover is the only simple explanation for $^{14}C$ concentrations below saturation in samples collected above 190 m, there is not an obvious relationship between proximity to present-day snowfields and whether a sample is $^{14}C$ saturated or not. The only exception to this is the highest sample from Mt. Seelig, which has a $^{14}C$ concentration below saturation and was collected from a very small outcrop (few square meters) that protrudes through the margin of the summit ice cap (Fig. 4c). All other samples from Mt. Seelig are estimated to have been collected within approximately 5-20 m of snowfields. How susceptible each sample site is to snow cover is likely related to local wind effects near the cliff edge, which are difficult to predict. Therefore the absence of a clear relationship between snowfield proximity and whether a sample is $^{14}C$ saturated is not surprising."*

- The authors cite unpublished Al/Be/Ne data to support their argument for low erosion rates and long exposure of bedrock surfaces in the Whitmore Mtns. The authors should either publish this data within this manuscript, or remove reference to them.

*The reference to unpublished data has been removed.*

- Comment on Section 5.1: the argument for simple monotonic post-LGM thinning of ice sheet margins despite a deglacial increase in snowfall is likely correct, but is overly simplified. In the Heritage Range, the competition between increased snowfall and dynamic thinning is argued to explain the delayed deglaciation at that site, which initiated in earnest only after ~10 ka when dynamic thinning began to outpace deglacial increases in snowfall (Hein et al., 2016). If correct, the influence of increased snowfall is more widespread than implied (i.e., it's influence extends beyond the divide), even though it may not have caused local thickening. Similar competition between processes has been suggested to explain ice histories in the Ross Sea sector (Hall et al., 2015; Todd et al., 2010).

*At the Pirrit Hills, the WAIS does appear to have thinned monotonically. The highstand was reached by the time that the accumulation rate in West Antarctica, as recorded at the WAIS Divide ice core site, began to increase from it's ice-age low (see Figure 5a of manuscript). This implies that thickening did not occur at the Pirrit Hills in response the accumulation-rate rise. As suggested by the reviewer, it is certainly possible that increased snowfall rates delayed the onset of thinning at the Pirrit Hills. To account for this, we have added the following sentence to the first paragraph of Section 5.1: "As discussed by Hein et al. (2016) in regard to the Heritage Range, the onset of thinning at the Pirrit Hills (and other sites) may have been delayed by the deglacial snowfall increase." We have also added a sentence to this effect to the end of the third paragraph of Section 5.2.*

Technical corrections:

- What is the distance of Pirrit Hills from the grounding line and the divide?

*The Pirrit Hills are ~200 km from both the grounding line and from the divide. The first sentence of Section 2.1 has been modified to convey this.*

-p3 line 30- statement that no glacially-transported cobbles found contrasts with next statement on discovery of indurated till. Rewrite to clarify.

*We appreciate the suggestion, but the first sentence in question does not state that no glacially-transported cobbles were found. Rather, it states that "we found no glacially-transported cobbles or boulders perched on bedrock surfaces." This is an important difference. One of the goals of these sentences is to contrast debris "perched on bedrock surfaces" with "indurated and weathered till". We have opted to not change the manuscript text.*

- P4L8 – replace "remove" with "minimize"

*Changed.*

-P4L15 – Perhaps include in Table the distance of each sample to the present snow/ice cap and reference here.

*See response to previous comment on snowfield proximity.*

-P4L19 –For clarity, state from where the ice margin elevation was determined at each site (given this can vary significantly around a nunatak).

*We have changed the text to indicate that sample heights are measured relative to the blue-ice areas that are at the base of each ridge.*

- P8L10 – can't use unpublished data to support your argument.

*The reference to unpublished data has been removed.*

- P9L35 – spelling of "ice"

*Fixed.*
* * *
**Additional changes**

In the abstract, we replaced the sentence "At the Pirrit Hills, ice reached a highstand ~320 m above present during the last glacial period." with "At the Pirrit Hills, evidence of glacial-stage ice cover extends ~320 m above the present ice surface." The purpose of this is because we cannot fully rule out the possibility that the highstand during the last glacial period was not somewhat higher than the highest deposits that we found.

Section 2.2 was accidentally labeled 2.1.1. This has been fixed.

**Author responses to the Editor's comments on "Thickness of the divide and flank of the West Antarctic Ice Sheet through the last deglaciation"**

Perry Spector et al.
Pspector@bgc.org

The Editors comments and our responses are shown in blue and balck, respectively. At the end of this document we also provide point-by-point documentation of additional, unsolicited changes we have made to improve the clarity of the manuscript.
* * *
**_Responses to the Editor_**

I would like to thank both reviewers for their detailed and constructive comments on this manuscript and also the authors for posting their response to the reviewers' comments.

Several issues are not fully resolved by the authors' response to the reviewers' comments. I note these below, along with a few additional points that the authors may want to consider as they prepare a revised version of their manuscript.

Reviewer 1 raises some concerns that have potential implications for the reliability of the paper's conclusions. I share some of these concerns, and in particular I feel that you do not always provide a full discussion of the uncertainties on the data or explore alternative explanations for the evidence. Some of the assumptions you make when interpreting the data have significant implications for the exposure/burial history that you subsequently infer. I encourage you to acknowledge the limitations of the data more thoroughly and discuss the viability of alternative scenarios where relevant.

In general, both reviews are positive, and they highlight the novelty and importance of this study. I therefore encourage you to submit a revised manuscript that addresses the points mentioned below and in the individual reviews.

Kind regards,

Pippa Whitehouse

We thank Pippa Whitehouse for a thorough review of the manuscript and for providing constructive comments.

Specific points

Both reviewers comment on the reliability of the saturated ages reported on page 7, lines 7-9. You provide an extensive discussion on this point in your response to the reviewers' comments, but you also state that you do not plan to include any additional information in the manuscript. Given that both reviewers commented on this, and given the importance of these ages in determining the ice history at Mt Seelig, I think it is important to include a brief discussion on the reliability of the saturated ages in the main text - if only for the benefit of those readers who are unfamiliar with the issues associated with analysing and interpreting such samples. As you note, the full details can remain in the review documents.

We agree that this issue should be mentioned in the manuscript. Previously we had found it challenging to figure out how and where to discuss it. We have now added a paragraph to Section 3.3 discussing this issue, and we refer readers to the interactive online discussion of this article for a more detailed discussion of the issue.

In two places Reviewer 2 mentions a number of earlier articles that discuss the competing roles of precipitation change and grounding line dynamics in controlling post-glacial Antarctic change. Of the articles mentioned by the reviewer, all are already cited in the original version of your manuscript except Hall et al. (2015, Nature Geoscience). I encourage you to consider including a reference to this highly relevant article in the revised version of your manuscript.

We have added reference to Hall et al. (2015) in Section 5.1.

Reviewer 1 queried whether there is evidence of atmospheric temperatures being warm enough to induce thinning across West Antarctica. Much of your response (and the text in the manuscript) appears to rely on the assumption that there is a delay of 10-30 kyr between atmospheric warming and ice thinning. You include a reference to an entire textbook, which makes it difficult to determine the precise basis for this assumption, but the text on line 4 (page 2) suggests that it relates to the time required for surface warming to have an impact on conditions at the base of the ice sheet. However, on lines 2-3 (page 2) you also mention the process by which an increase in ice temperature (at any depth) will change the rheology of the ice, thus allowing it to deform and flow more easily. The time lag for this second process is presumably much shorter, perhaps negating your assumption that there must be a delay of at least 10 kyr between warming and thinning? And in fact, I don't think the reviewer is even asking whether warming-induced thinning has commenced, but rather whether the deglacial increase in atmospheric temperature was sufficient to trigger thinning by one of the processes described above. Please address this second point.

We agree that referencing an entire textbook was not particularly useful. In each case we were referring to Chapter 11.4.2 of Cuffey & Patterson (2010), which discusses the response of ice sheets at the end of an ice age. We have changed these citations to include the specific chapter.

As you mention, warming the ice at any depth will change its rheology and allow it to flow faster. However, because the bulk of ice deformation occurs near the base of the ice sheet, most of the effect of surface warming on ice-sheet thickness does not occur until that warming has propagated to near the base of the ice sheet. As discussed in Cuffey & Patterson (2010, Ch. 11.4.2), this is expected to require roughly 10 to 30 kyr.

In the previous draft of the manuscript, we had mentioned this thinning mechanism in the introduction, but then not discussed it further in the paper. This is because, given the time delay of this mechanism, along with the fact that surface temperatures remained relatively low until ~15 kyr B.P. (Cuffey et al., 2016, PNAS), it appeared unlikely that this mechanism could have significantly contributed to thinning in West Antarctica. In the revised manuscript, we acknowledge the possibility that some thinning by this mechanism could have occurred in the late Holocene. Mention of this now appears in Section 1 and Section 5.1. This does not actually affect our conclusions. At the Pirrit Hills, any thinning by this mechanism would likely have occurred only after the majority of thinning to the modern ice level was complete. At the Whitmore Mountains, the possibility of late Holocene thinning by this mechanism does not change the fact that, as stated in the text, "thinning to the modern ice level at Mt. Seelig could not have occurred before 7 kyr ago (i.e. before modern ice levels were reached on lower Reedy Glacier)".

Opening sentence of section 5.1: "…despite the deglacial increase in snowfall…" It is not clear what evidence you are drawing on to support this statement, but elsewhere in the manuscript I note that you refer to the WAIS Divide ice core when discussing accumulation change across West Antarctica. The Pirrit Hills are in a different catchment to the WAIS Divide ice core (figure 1 of your manuscript) and hence they may have experienced a different snowfall history to that at WAIS Divide (page 10, line 24 of your manuscript). The statement at the start of section 5.1 therefore requires additional justification if you wish to use accumulation change at WAIS Divide as a proxy for accumulation change at the Pirrit Hills. If you are drawing on alternative evidence to support the statement about accumulation change at the Pirrit Hills then please make this clear. In light of my comments, please also check the robustness of the statement in the conclusions that refers to accumulation rates at the Pirrit Hills.

We agree that our justification for this sentence was not obvious. We had discussed this issue regarding the Whitmore Mountains, and the reasoning is the same for the Pirrit Hills. For both sites, the magnitude of accumulation-rate changes may not have been the same as at WAIS Divide, but the timing of changes were probably similar because (i) all three sites are fed by storms originating in the Amundsen Sea low, and (ii) the accumulation rate increased considerably in both East and West Antarctica at this time. To address this issue, we have moved this discussion (for both sites) to the beginning of Section 5.1.

On page 9, you draw on evidence from sites across West Antarctica to support your inference that ice was previously thicker in the Whitmore Mountains. Considering the likely flowlines of the

ice sheet during the last deglacial period, it is not clear to me that ice thickness changes at Mt Waesche (page 9, line 16) should necessarily be similar to those at the Whitmore Mountains. Similarly, one could envisage a scenario whereby ice was thicker than present at Byrd Station during the LGM (page 9, line 19) but not at the divide upstream of this site. It would be useful if you could include a statement about the degree to which ice thickness changes at Mt Waesche and Byrd Station can be expected to co-vary with ice thickness at the Whitmore Mountains (as you do when discussing evidence from the Ohio Range).

We agree that ice-thickness changes at Mt. Waesche are difficult to connect to changes at the Whitmore Mountains, and we have removed reference to Mt. Waesche here. Reviewer 1 strongly suggested that we not discuss the behavior of ice-sheet models in this section because later in the paper we evaluate those same models. We agreed that this was a good point, and we complied with the recommendation. Therefore, with regard to the evidence of thickness changes from Byrd Station, it is difficult to provide a quantitative statement about the degree to which thickness changes are expected to co-vary at this site and at the Whitmore Mountains, as you recommended.

Page 9, line 33: "Thinning to the modern ice level at Mt. Seelig therefore could not have occurred before 7 kyr ago". To improve the clarity of your argument, please be more explicit about which of the constraints mentioned in the previous paragraph you are drawing on to make this quantified statement.

We agree that this sentence was not clear. It now reads "Thinning to the modern ice level at Mt. Seelig therefore could not have occurred before 7 kyr ago (i.e., before modern ice levels were reached on lower Reedy Glacier)."

Page 10, line 10: could the ice have been thicker than present for a brief period during the LGM? i.e. could it be that the samples were not completely saturated at the beginning of the ~15ka burial period?

This is a good question. The model we use to investigate the possible histories of exposure and ice cover (equation 1) assumes that samples were initially $^{14}$C saturated. Relaxing this assumption actually restricts the set of exposure and ice-cover histories that are permitted by the observations. So, if the two lowest elevation samples from Mt. Seelig were not initially saturated, then the onset and duration of ice cover would be later and more brief, respectively, than implied by Figure 6a. To give an example, if we assume that the samples began with C-14 concentrations 5% below saturation, it implies that the highstand was reached within the past ~13 kyr as opposed to the past ~15 kyr. This would only strengthen our findings that ice cover was relatively recent and brief.

In the previous draft of the manuscript, the effect of this assumption (i.e., initial saturation) on our findings was not explained well. We have rewritten parts of Section 4.3 because we did not think that our explanations of the C-14 constraints and of Figure 6a were sufficiently clear. In

addition, we have also added a paragraph discussing the assumption of initial C-14 saturation and what effect relaxing this assumption would have. This issue is also discussed some in Section 5.1.

Page 10, line 31: ICE-6G is not really a 'model of glacial isostatic adjustment'; it is an ice history model in the sense you are using it

We now refer to ICE-6G as a model of ice-sheet history.

Please include latitude and longitude labels on figure 1

This issue was commented on previously, and we did add labels to the figure at that time. The labels are purposefully subtle, so as to not distract from the rest of the figure, but we believe that they are sufficiently visible and legible.
* * *
**Additional Changes**

Section 1: It was previously somewhat vague whether the second paragraph of the Introduction was referring to ice sheets generally or to the WAIS specifically. We have reworded parts of this paragraph to make it clear that we are talking about the behavior of the WAIS at the end of the last ice age. Other changes to the Introduction are either (i) discussed above in our response to the Editor or (ii) largely stylistic.

Section 2.1: We made a few very minor stylistic changes to this section.

Section 3.3: We have made the following changes to this section:
*Paragraph 1*
- We clarified the description of our analytical methods for extracting Be from quartz.
- We expanded the discussion about sample contamination by beryl or other Be-bearing minerals. The text now explicitly states that we calculated $^{10}$Be concentrations using ICP-OES determinations of total Be rather than the amount of Be added as carrier.
- We expanded the description of our $^{10}$Be blanks and our blank corrections.
*Paragraph 2*
- We have provided a more full account of the scatter and bias in Be isotope ratio measurements from LLNL, and how we have increased the uncertainty of our sample Be isotope ratios to account for this.

Section 4.1: We made minor stylistic changes to the third paragraph of this section. In the fourth paragraph, we have changed the first and last sentences so that the paragraph now describes exposure-dating results in the Weddell Sea sector before drawing inferences about them.

Section 4.2: We made minor changes to this section to (i) make it easier to read and (ii) better explain the relationship between sample sites and present-day snowfields.

Section 4.3: Previously, the explanation of the model we use to interpret the $^{14}$C results was somewhat confusing. We have rewritten much of Section 4.3, and we believe that it is now much clearer, and that it makes Figure 6a easier to understand as well.

Section 5.1: Various changes to this section are described above in our response to the Editor. Additionally, we have combined the last two paragraphs of this section, which makes it easier to read.

Section 5.2: We have made a minor stylistic change toward the end of this section.

**Author responses to the Editor's comments on "Thickness of the divide and flank of the West Antarctic Ice Sheet through the last deglaciation"**

Perry Spector et al.
Pspector@bgc.org

The Editor's comments and our responses are shown in blue and balck, respectively.
* * *
***Responses to the Editor***

I would like to thank the authors for addressing all comments on the previous version of the manuscript. In particular, the interpretation of your results is much clearer and your overall conclusions are more robustly justified.

We thank the editor for another thorough review.

In light of your edits, there are two points that require additional clarification:

1) Page 10, lines 26-27: you mention that your results are based on the assumption that the samples close to the present ice surface have "experienced only one period of exposure during the past ~30 kyr". This implies that you assume the samples were previously covered, but the timing or duration of this coverage is not clear. Please address this. In addition, in light of the text on lines 24-25 please make it clear which "pair of samples" you are referring to on line 26, and if appropriate adopt the terminology introduced above that refers to a "simple exposure age".

I see that this was worded in a confusing way. The first few sentences of the paragraph now read as follows:

> The $^{14}C$ concentrations of the two samples collected near the modern ice surface would correspond to simple exposure ages of ~10-15 kyr, under the assumption that they were previously ice-covered for a sufficient amount of time to remove any pre-existing $^{14}C$. This scenario is represented by the asymptote of the "zero snow cover" band in Figure 6a. However, under the three-stage model of Equation 1, this pair of samples could have also been exposed by WAIS thinning (i) within the past ~10 kyr, with more recent re-exposure requiring relatively brief prior ice cover, or (ii) prior to ~15 kyr B.P., given the possibility of cover by local snow fields (Fig. 6a).

2) Page 12, lines 34-35: In the previous paragraph you mention that it is possible the samples were not 14C-saturated when they were last covered, i.e. they could have been briefly covered during the LGM, then uncovered, and covered again at 15 ka BP. Given this possibility, please re-consider the robustness of your statement that the "constraints demonstrate that the WAIS at the Whitmore Mountains was the same thickness or thinner than present during the LGM".

Please also clarify what time period you are referring to when you talk about the LGM (apologies if I have missed this).

We agree that this was confusing. We have changed the sentence to read: "These constraints demonstrate that the WAIS at the Whitmore Mountains was the same thickness or thinner than present prior to the most recent highstand, and that this highstand was reached sometime in the last ~15 kyr."

I list below a number of minor technical points that also require attention. Once these and the issues above are addressed, I would be happy to review a revised version of the manuscript.

Kind regards,

Pippa Whitehouse

Minor technical points (suggested edits in **bold**)

Page 2, line 29: delete 'also' as there is no previous mention that data are scarce
We believe that 'also scarce' is appropriate here because the prior sentence begins with "The only relevant constraints…", which implies a scarcity of data.

Page 3, line 17: 'isolated **glacial** deposits'
Fixed.

Page 6, line 12-13: 'of **the** UW standards'
Fixed.

Page 8, line 13: 'was **at** least'
Fixed.

page 10, line 25: revise back to '130 m **above the ice surface**'
Fixed.

Page 10, line 31: delete one instance of 'our' and edit 'initial' -> 'initially'
Fixed.

Page 11, line 19: clarify what type of deformation you are referring to, presumably ice
Clarified that we are referring to ice deformation here.

Page 11, line 22: the reference to Hall et al. (2015) seems a little out of place in this sentence that specifically talks about the Heritage Range and Pirrit Hills. It may sit more appropriately with the other references listed on page 2, line 19

We agree that this reference would be more appropriate in the introduction. Fixed.

Page 12, line 28/fig. 6b: in line with an earlier edit, remove references to Mt. Waesche

We have removed reference to Mt. Waesche in Section 5.1. However, we have kept Mt. Waesche in Figure 6b to provide the reader with more information about ice thickness changes across West Antarctica. This is analogous to including the Hudson Mountains and the Ford Ranges in Figure 6d, which provide the reader with the range of times when sites across West Antarctica reached present-day ice levels even though there is no expectation that these wo sites share similar ice-thickness histories with the Whitmore Mountains.

Figure 2: refer to (a) and (b) in the first sentence of the caption

Good catch. Fixed.

Figure 7: add labels (a) to (d) on appropriate panels

Another good catch. Fixed.

Additionally, we made the following changes:

1. First paragraph of Section 3.3: Changed the beginning of the sentence from "One batch of samples…" to "The batch of samples…".
2. Corrected a typo in the caption of Figure 6.
3. We have also added '$t_{cover}$' and '$t_{expose}$' to the axis labels of Figure 6a.

[revised manuscript text omitted]